# Effects of Alternating Irrigation with Fresh and Saline Water on the Soil Salt, Soil Nutrients, and Yield of Tomatoes

**Jingang Li [1], Jing Chen [2,*], Zhongyi Qu [3,*], Shaoli Wang [4], Pingru He [5] and Na Zhang [6]**

1   College of Water Conservancy and Hydropower Engineering, Hohai University, Nanjing 210098, China
2   College of Agricultural Engineering, Hohai University, Nanjing 210098, China
3   Water Conservancy and Civil Engineering College, Inner Mongolia Agricultural University,
    Hohhot 010018, China
4   State Key Laboratory of Simulation and Regulation of Water Cycle in River Basin, Department of Irrigation
    and Drainage, China Institute of Water Resources and Hydropower Research, Beijing 100038, China
5   Key Laboratory of Agricultural Soil and Water Engineering in Arid and Semiarid Areas of Ministry of
    Education, Northwest A&F University, Yangling 712100, China
6   Ningxia Institute of Water Resources Research, Yinchuan 750021, China
*   Correspondence: chinsei@163.com (J.C.); quzhongyi@imau.edu.cn (Z.Q.);
    Tel.: +86-139-1298-0055 (J.C.); +86-150-4910-9708 (Z.Q.)

**Abstract:** Saline water irrigation has become extremely important in arid and semi-arid areas in northwestern China. To study the effect of alternating irrigation models on the soil nutrients, soil salts, and yield of tomatoes with fresh water (total dissolved solids of $0.50$ g·L$^{-1}$) and saline water (total dissolved solids of $3.01$ g·L$^{-1}$), a two-year field experiment was carried out for tomatoes in the Hetao Irrigation District (HID), containing six drip irrigation models: T1 (all freshwater irrigation), T2 (saline water used in the seedling and flowering stages; fresh water in the fruit-set and breaker stages), T3 (saline water in the flowering and fruit-set stages; fresh water in the seedling and breaker stages), T4 (saline water in the fruit-set and breaker stages; fresh water in the seedling and flowering stages), T5 (saline water in the flowering and breaker stages; fresh water in the seedling and fruit-set stages), T6 (saline water in the seedling and fruit-set stages; fresh water in the flowering and breaker stages). The study found that saline water irrigation tends to have a positive effect on soil total nitrogen and a negative influence on soil total phosphorus at each growth stage of the tomato. Soil Na$^+$, Mg$^{2+}$, Ca$^{2+}$, K$^+$, and Cl$^-$ increased over the growth period, soil HCO$_3$$^-$ decreased gradually by growth stage, and the salt ions increased with the amount of saline water applied in alternating irrigation. Though the soil salt accumulated in all experimentally designed alternating irrigation models, soil alkalization did not occur in the tomato root zone under the soil matric potential threshold of $-25$ kPa. The utilization of saline water resulted in about a 1.9–18.2% decline in fruit yield, but the total soluble solids, lycopene, and sugar in the tomato fruits increased. Ultimately, drip irrigation with fresh water at the seedling to flowering stages and saline water at the fruit-set to breaker stages was suggested for tomato cultivation in HID.

**Keywords:** Hetao Irrigation District; alternating irrigation; saline water; tomato; mulched drip irrigation

## 1. Introduction

Due to low rainfall and intense evaporation, the Hetao Irrigation District (HID) mainly relies on Yellow River water to meet its agricultural water demand. However, with the development of

the economy and national regulations on ecology in recent years, the amount of available Yellow River water is gradually decreasing, which exacerbates the gap between the high consumption rate of farmland and the freshwater shortage. To alleviate the shortage of fresh water, it is critical to find a balance between the rising water demands and available water for agriculture, especially in arid and semi-arid areas such as HID. There is an urgent need for the extension and application of water-saving irrigation measures, as well as the exploitation of unconventional water resources. Applying alternating water resources for irrigation in agriculture has become one of the important patterns to alleviate the freshwater shortage. As one of the major unconventional water resources, saline water has been extensively used for agricultural irrigation for hundreds of years [1–3].

Shallow saline ground water (buried depth range: 0–40 m), which is plentiful in HID, is usually characterized by the average total dissolved solids (TDS) of 2.54 $g \cdot L^{-1}$ [4]. The allowable yield of shallow groundwater is 1.66 billion $m^3$, while the available brackish water ($2.0 \leq TDS \leq 3.0\ g \cdot L^{-1}$) reserves 0.721 billion $m^3$ [5]. Studies on saline water irrigation have found that, on the one hand, there are many favorable impacts, such as alleviating freshwater shortages, accelerating groundwater renewal, reducing soil salt accumulation in dry seasons, and promoting soil salt desalination in rainy seasons [6–8]; on the other hand, there are some potential hazards, such as introducing more salt into the farmland, which may result in salt accumulation—as a result, the soil environment of farmland and the direction of soil water-salt movement was changed, and ultimately crop water uptake was hindered [9–12]. Thus, it is ecologically important to study the proper irrigation models for saline water irrigation.

The traditional flood irrigation method in HID applies large amounts of water, which results in a rise in the groundwater level and a high salt concentration in the root zone. In order to control soil salinity, excess water was applied to the farmland, which trapped it in a vicious cycle. In recent years, mulched drip irrigation was supposed to be the most effective method of saline water irrigation due to the characteristics of distributing water and nutrients uniformly, controlling the amount of applied water precisely at high frequencies, reducing evaporation by plastic mulch, minimizing deep percolation with normal irrigation quota, and decreasing the adverse effects of salinity by means of leaching [13–15].

However, the ability of roots to take in water and nutrients was dramatically damaged with the long-term saline water irrigation, so effective models for saline water irrigation to alleviate the negative effects of soil salt on plants and soil health are needed. There are three main approaches to utilizing saline water: all saline water irrigation, which usually leads to soil salt accumulation and crop yield reduction; fresh-saline water mixture irrigation, which keeps the salinity of irrigation water lower than the threshold of the target crop by mixing saline and fresh water to reach a relatively low level of salinity [16–18]; and alternating fresh-saline water irrigation, which irrigates crops alternately with fresh and saline water [19]. Though fresh-saline water mixture irrigation was proven to be an useful method for saline water utilization, there are still limitations to its effectiveness, such as reservoirs being necessary for mixing the two irrigation water sources [20], while the alternating irrigation method with fresh-saline water was suggested to control the topsoil salt in drip irrigation systems. Murad et al. (2018) demonstrated that irrigation with fresh water at early sensitive stages, combined with saline water at later, more tolerant stages, can minimize the yield loss of maize in saline coastal regions of Bangladesh [8]. With the combined use of saline drainage water and freshwater irrigation, Sharma et al. (1994, 2005) indicated that, though the crop yield decreased by 6–18%, there was no significant soil salt accumulation [21,22]. Malash et al. (2005) reported that alternating fresh-saline water irrigation contributes to decreasing the damage of salt to crops and soils [23].

As a crucial vegetable crop around the world, the tomato (*Lycopersicon esculentum*) is widely cultivated in HID, where the suitable environment and sunshine conditions are beneficial to the accumulation of sugar. Tomato lycopene is considered to have a positive impact in the prevention of atherosclerosis and some cardiovascular diseases [24]. Though the tomato is considered to be a medium salt-sensitive crop [25,26], supplementary irrigation with brackish water during freshwater

shortage was necessary [27,28]. Wang et al. (2007) conducted a field experiment about mulched drip irrigation with saline water at a conductivity level of 4.2 dS·m$^{-1}$, and found that there was no soil salt accumulation during the growth period of the tomato [29]. Karlberg et al. (2007) found that the fruit production of tomatoes with saline water drip irrigation was above the average yield [30]. Wan et al. (2008) have carried out an irrigation experiment with water containing different concentrations of salt for three consecutive years (2003–2005); the results implied that saline water drip irrigation with a salt concentration of 1.1–4.9 dS·m$^{-1}$ has little influence on the root length density, maximum leaf area index (LAI), total chlorophyll content, and fruit yield [31].

Though there are many studies on the impacts of alternating irrigation on the yield and quality of tomatoes, cotton, and lemons, and on the formation of clogging substances inside the emitters [32], few studies have been found to establish an alternating irrigation schedule based on growth stages under mulched drip irrigation. The current experiment was conducted to: (1) estimate the influence of different alternating irrigation models on the total nutrients and available nutrients; (2) explore the effect of different alternating irrigation models on soil salt ions and the sodium adsorption ratio (SAR); and (3) investigate soil salinization under the use of different water resources.

## 2. Materials and Methods

### 2.1. Experimental Conditions

The field experiment was carried out in 2017-2018 at the Jiuzhuang Experimental Station, in HID, China (longitude: 107°18′ E, latitude: 40°41′ N 1042 m a.s.l.). According to the meteorological data, which were collected from an automatic weather station (YM-03A, Handan Yimeng Electronics Co., Ltd., Hebei province, China) installed 50 m away from the field experiment site, the average annual temperature at the experimental area was 6.8 °C, and the area is characterized as having a mid-temperate semiarid continental climate with little precipitation (annual average of 140 mm) and large evaporation (annual average pan evaporation exceeding 2032–3179 mm) [33]. The total precipitation was 142 mm and 131 mm in 2017 and 2018, respectively, and mostly occurred from June to August, while the reference crop evaporation and precipitation during the growing period of the tomato are shown in Figure 1.

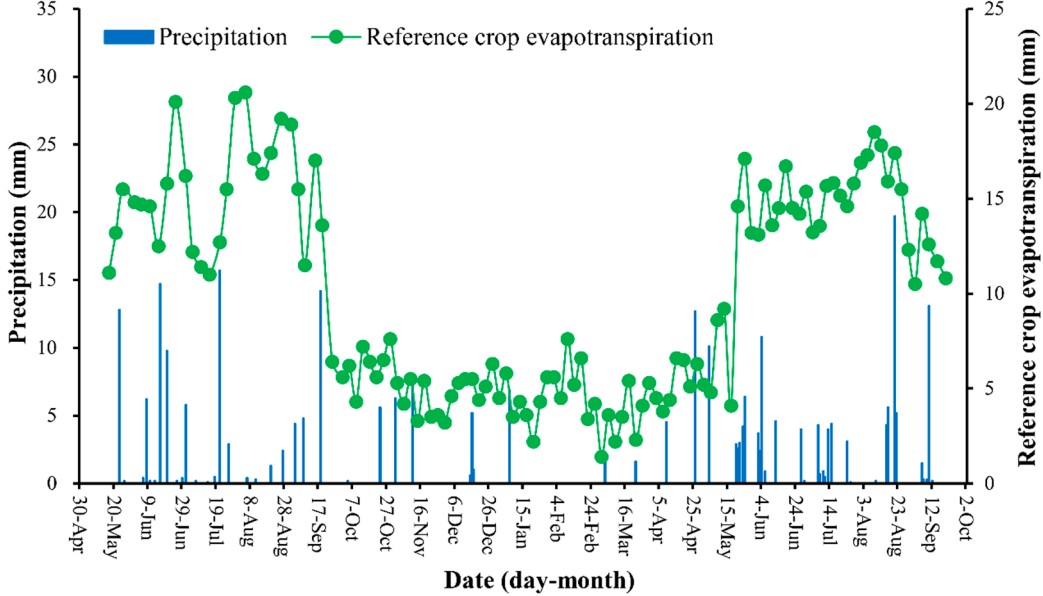

**Figure 1.** The reference crop evapotranspiration and precipitation in 2017 and 2018.

On 9 April 2017, 25 sampling locations were selected randomly in the experimental field, and the samples were collected at 0–20 cm, 20–40 cm, 40–60 cm, 60–80 cm, and 80–100 cm depth; the soil

physical properties, nutrient contents, and salt ions contents were also measured. According to the soil sample analysis, the soil 100 cm underneath the surface at the experiment field can be divided into three layers on the basis of an international soil texture classification system. The detailed physical properties and background values of soil nutrients and soil salt ions are shown in Tables 1–3, respectively.

**Table 1.** Soil physical properties in the experimental area.

| Soil Depth (cm) | Bulk Density (g·cm$^{-3}$) | Field Capacity (%) | Porosity (%) | Silt (%) | Sand (%) | Clay (%) | Soil Salinity (g·kg$^{-1}$) | Texture Class |
|---|---|---|---|---|---|---|---|---|
| 0–20 | 1.39 | 23.7 | 48.03 | 24.31 | 62.09 | 13.60 | 1.317 | Sandy loam |
| 20–40 | 1.42 | 23.3 | 44.32 | 25.64 | 60.25 | 14.11 | 0.868 | Sandy loam |
| 40–60 | 1.37 | 25.4 | 46.54 | 18.65 | 61.18 | 20.17 | 0.744 | Sandy loam |
| 60–80 | 1.54 | 14.9 | 36.45 | 87.78 | 11.16 | 1.06 | 1.263 | Sand |
| 80–100 | 1.43 | 19.2 | 42.84 | 32.57 | 51.28 | 16.15 | 1.845 | Middle loam |

**Table 2.** Soil nutrient contents in the experimental area.

| Soil Depth (cm) | Total Nutrient Contents (g·kg$^{-1}$) | | | Available Nutrient Contents (mg·kg$^{-1}$) | | | |
|---|---|---|---|---|---|---|---|
| | Total N | Total P | Total K | Ammonium N | Nitrate N | Available P | Available K |
| 0–20 | 1.169 | 0.913 | 17.115 | 74.615 | 113.250 | 12.625 | 291.25 |
| 20–40 | 0.963 | 0.818 | 18.490 | 51.925 | 16.885 | 9.800 | 318.50 |
| 40–60 | 0.861 | 0.668 | 19.590 | 76.930 | 12.960 | 4.025 | 186.00 |
| 60–80 | 0.477 | 0.595 | 16.150 | 40.120 | 7.850 | 3.350 | 123.50 |
| 80–100 | 0.368 | 0.577 | 13.740 | 30.860 | 3.120 | 5.800 | 100.50 |

**Table 3.** Soil salt ions content of the test field.

| Soil Depth (cm) | Salt Ions Content (mmol·L$^{-1}$) | | | | | |
|---|---|---|---|---|---|---|
| | $HCO_3^-$ | $Cl^-$ | $SO_4^{2-}$ | $Ca^{2+}$ | $Mg^{2+}$ | $K^+ + Na^+$ |
| 0–20 | 0.643 | 4.628 | 3.145 | 1.367 | 1.139 | 5.910 |
| 20–40 | 0.610 | 2.739 | 2.784 | 1.256 | 1.072 | 3.805 |
| 40–60 | 0.587 | 2.433 | 3.433 | 1.467 | 1.611 | 3.487 |
| 60–80 | 0.729 | 1.517 | 2.106 | 1.233 | 0.567 | 2.551 |
| 80–100 | 0.686 | 1.450 | 2.056 | 1.317 | 0.511 | 2.363 |

The groundwater buried depth in the area varies from 1.85 m to 3.68 m below the soil surface during the growth period of the tomato, which is shown in Figure 2. The average groundwater depth during the tomato growth period was 2.510 m in 2017 and 2.46 m in 2018, respectively. The traditional flood irrigation water is mainly introduced from the Yellow River, which has an annual average salt concentration of 0.505 g·L$^{-1}$ (electrical conductivity of 0.79 dS·m$^{-1}$), while the average annual salinity of the local shallow groundwater is 3.006 g·L$^{-1}$ (electrical conductivity of 4.70 dS·m$^{-1}$) (Figure 2). The main eight ion contents, pH, and TDS of shallow groundwater and Yellow River water are shown in Table 4.

**Table 4.** The features of Yellow River and shallow groundwater.

| Characteristics | Shallow Groundwater | Yellow River Water | Characteristics | Shallow Groundwater | Yellow River Water |
|---|---|---|---|---|---|
| $Ca^{2+}$ (mg·L$^{-1}$) | 186.00 | 50.00 | $Cl^-$ (mg·L$^{-1}$) | 798.50 | 138.45 |
| $Mg^{2+}$ (mg·L$^{-1}$) | 253.70 | 26.00 | $SO_4^{2-}$ (mg·L$^{-1}$) | 285.50 | 79.20 |
| $K^+$ or $Na^+$ (mg·L$^{-1}$) | 986.00 | 108.10 | TDS (g·L$^{-1}$) | 3.01 | 0.50 |
| $HCO_3^-$ (mg·L$^{-1}$) | 496.00 | 330.40 | pH | 7.46 | 7.38 |
| $CO_3^2$ (mg·L$^{-1}$) | 0.00 | 0.00 | | | |

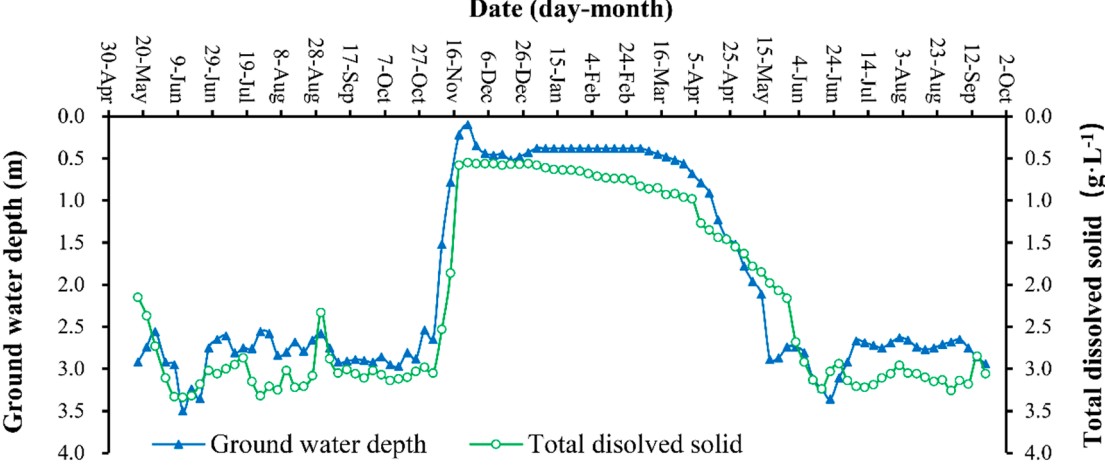

**Figure 2.** Buried depth and TDS of shallow groundwater in 2017 and 2018.

## 2.2. Experimental Design

A local variety of tomato (Jinye No.1) was selected as the test crop in the current study; its growth can be divided into five stages according to the growth and reproductive characteristics of the tomato: seedling stage, flowering stage, fruit-set stage, breaker stage, and maturity stage. Based on the five growth stages, five alternating irrigation models with saline water (TDS of 3.0 g·L$^{-1}$) and fresh water (TDS of 0.5 g·L$^{-1}$) were built to a randomized complete block design. The tomatoes were planted in ridged raised beds (height 0.3 m × width 0.9 m × length 25 m) with plastic film over the roots (width 1.0 m × length 25 m) and drip tubes with 0.3 m emitter intervals were located in the center of each bed. Then 36 beds were divided into six plots for each treatment in one plot, and each plot occupied an area of 225 m$^2$ (width 7.9 m × length 25 m), as shown in Figures 3 and 4. The tomatoes were transplanted in two rows at a row spacing of 60 cm and plant spacing of 30 cm. The irrigated water amount was measured by a water meter installed at the inlet of the drip submain unit, which comprised six beds; each treatment was equipped with three vacuum gauge tensiometers installed 20 cm underneath the emitter for SMP monitoring, and the tensiometers were observed and the measurements recorded three times a day at 8:00, 14:00, and 18:00 during the growth period of the tomato.

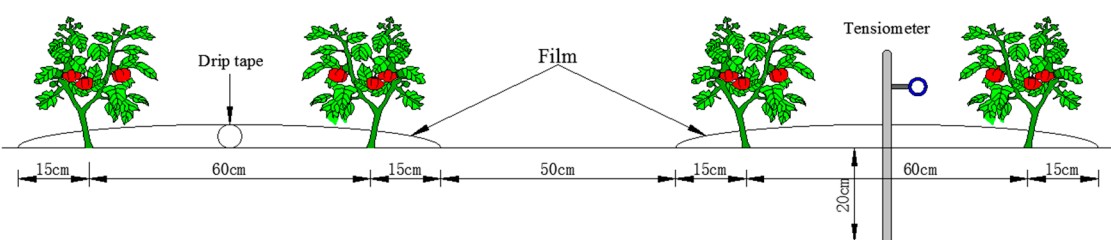

**Figure 3.** Sketch of drip lines and tensiometer.

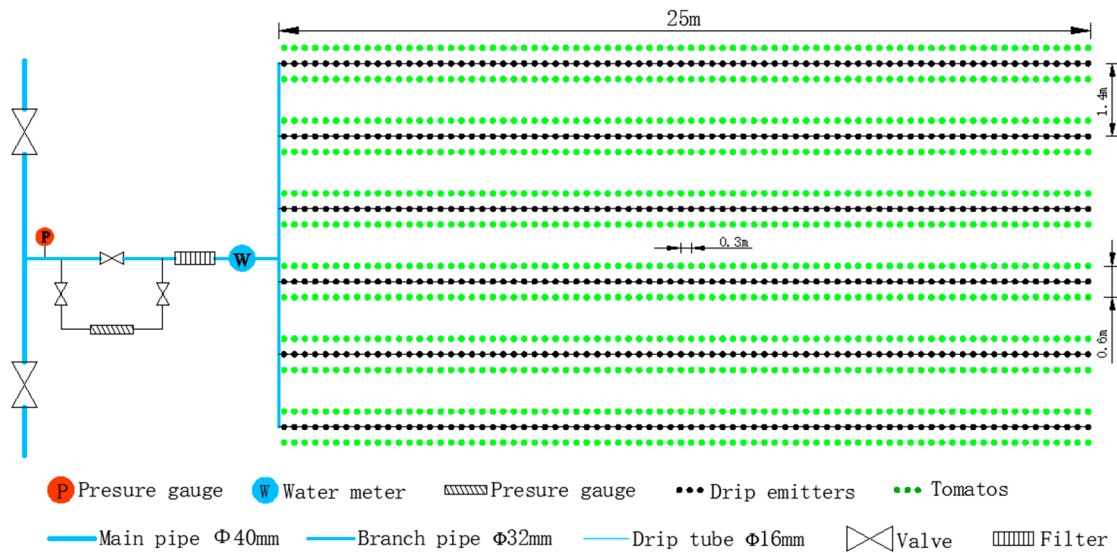

**Figure 4.** Layout of each plot.

### 2.3. Irrigation and Fertilization

According to the extensive investigation in 2016, traditional tomato cultivation is generally under film covering, while surface flood irrigation of 3600 $m^3 \cdot ha^{-1}$ in total was applied during the tomato growth period; additionally, supplementary irrigation of 900 $m^3 \cdot ha^{-1}$ was applied before transplanting to enhance the survival rate of the seedlings. Meanwhile, base fertilizers, including diammonium phosphate (DAP: 18% N, 46% P, 0% K) of 450 $kg \cdot ha^{-1}$ and potassium sulfate (including $K_2SO_4$, 45%) of 90 $kg \cdot ha^{-1}$ were uniformly applied to all plots before transplanting, with urea (46.2% N) of 180 $kg \cdot ha^{-1}$ given as a dressing fertilization.

The fertilization schedule of the present field experiment was based on the traditional fertilization schedule. Diammonium phosphate at a rate of 225 $kg \cdot ha^{-1}$ and potassium sulphate at a rate of 90 $kg \cdot ha^{-1}$ were applied as the base fertilization, and the dressing was supplied with urea (46% N) of 90 $kg \cdot ha^{-1}$ by mixing it with irrigation water at a concentration of 30% (*w/w*); the topdressing time and other agronomic practices were the same as in traditional tomato planting.

Many studies have indicated that the soil matric potential (SMP), which is measured 20 cm immediately underneath the drip emitter, can be used as an indicator for crop drip irrigation scheduling [34–36]. Kang et al. (2012) indicated that the SMP measured at 0.2 m underneath was recommended to be kept above −20 kPa when applying drip irrigation with saline water in Northwest China [37]. Fresh water with an amount of 900 $m^3 \cdot ha^{-1}$ was applied for supplementary irrigation before transplanting to leach the topsoil salt. Then, after the tomato seedlings' grafting (13 May 2017 and 2018), irrigation was triggered by tensiometers buried 0.2 m underneath the emitters; as soon as the value of all tensiometers in each treatment was lower than −25 kPa, irrigation was implemented for each treatment with corresponding irrigation water resources (Table 5), and the same irrigation amount, 200 $m^3 \cdot ha^{-1}$. Additionally, no irrigation water was applied to tomatoes at the ripening stage to avoid rotting.

**Table 5.** Treatments of tomatoes with saline water.

| Treatment | Seedling Stage | Flowering Stage | Fruit-Set Stage | Breaker Stage | Maturity Stage |
|-----------|----------------|-----------------|-----------------|---------------|----------------|
| T1 | FW | FW | FW | FW | |
| T2 | SW | SW | FW | FW | |
| T3 | FW | SW | SW | FW | |
| T4 | FW | FW | SW | SW | |
| T5 | FW | SW | FW | SW | |
| T6 | SW | FW | SW | FW | |

Note: "FW" is irrigation with fresh water (Yellow River water, TDS of 0.5 g·L$^{-1}$), while "SW" is irrigation with prepared saline water (shallow ground saline water, TDS of 3.0 g·L$^{-1}$).

### 2.4. Measurements

#### 2.4.1. Soil Salinity

Soil samples were taken during each growth period of the tomato at depths of 0–40 cm and 40–100 cm in the two locations, 0 cm (inside the film) and 75 cm (outside the film) from the drip tube, for the measurement of soil salinity. The soil samples were naturally air-dried and ground well to pass through a 1-mm sieve; then the soil leachates were prepared at a soil to water ratio of 1:5, $EC_{1:5}$ was measured by a conductivity meter (FE30, METTLER TOLEDO, Shanghai, China), and the pH value was determined by a pH meter (FE20, METTLER TOLEDO). Soil $Ca^{2+}$ and $Mg^{2+}$ were determined by an atomic absorption spectrophotometer (Varian spectra AA55, Varian, Palo Alto, California, USA), while soil $K^+$ and $Na^+$ were measured by a flame emission spectrophotometer (Model 410 Flame photometer, Sherwood Scientific, Ltd., New York, USA) [38]. Soil $Cl^-$ was measured by the method of silver nitrate titration, while soil $SO_4{}^{2-}$ was measured by the EDTA titration method; meanwhile, the soil $HCO_3{}^-$ was determined by the neutralization titration method.

SAR is usually selected as the index for measuring the level of soil salinization that results from irrigation models, and the SAR for soil samples was calculated using the following equation:

$$SAR = \frac{c(Na^+)}{\sqrt{\frac{c(Ca^{2+})+c(Mg^{2+})}{2}}}, \tag{1}$$

where $c(Na^+)$, $c(Ca^{2+})$, and $c(Mg^{2+})$ are the content of $Na^+$, $Ca^{2+}$, and $Mg^{2+}$, which are expressed in meq·L$^{-1}$.

The soil salinity of each soil layer was estimated using the soil salt content of samples both 0 cm and 75 cm away from the dripper. Soil salt concentration was estimated by electrical conductivity $EC_{1:5}$ via the following equation [39]:

$$Y = 0.349 \times EC_{1:5}, \tag{2}$$

where Y is the mass salt content (%) and $EC_{1:5}$ is the soil electrical conductivity (dS·m$^{-1}$).

#### 2.4.2. Soil Nutrient Content

The soil samples that were taken before transplanting and during each growth period were used for measuring the total nitrogen and total phosphorus. While the total nitrogen was measured by the Kjeldahl distillation method [40], the total phosphorus was determined following dry combustion at 550 °C for 2 h and extraction with 0.5 M $H_2SO_4$ [41].

#### 2.4.3. Fruit Yield and Quality

Tomato fruits were picked twice a week during the harvest seasons for each plot, and all fruits were classified as unmarketable (fruits that were cracked, green, sunburnt, with symptoms of blossom-end

rot, or damaged by pests) or marketable ones; the fruit yield was measured for the marketable fruits of each plot. The yield (g/plant) of each plant was determined during the harvest season.

For each treatment, 20 marketable tomato fruits were collected randomly for the measurement of total soluble solids ($T_{SS}$), lycopene, total sugar ($T_s$), and total acid ($T_a$). $T_{SS}$ was measured by an ACT-1E digital refractometer (ATAGO Co., Ltd., Tokyo, Japan), and the content of lycopene was measured using the spectrophotometry method. $T_s$ was determined by the Fehling reagent titration method, and $T_a$ was measured by the sodium hydroxide titration method.

### 2.5. Statistical Analysis

The soil nutrient, soil ions, and soil salinity were analyzed with Excel 2016 (Microsoft Corporation, Redmond, Washington, USA), while the layout of the drip lines was drawn by AutoCAD 2018 (Autodesk Inc., San Rafael, California, USA).

## 3. Results

### 3.1. Irrigation

Due to the variable sensitivity of tomato roots to soil moisture and soil salinity at different growth stages, the amount of water consumed and the irrigation frequency were usually distinct between models at different growth periods. Details on the amount of water applied according to the design in 2017 and 2018 are shown in Figure 5.

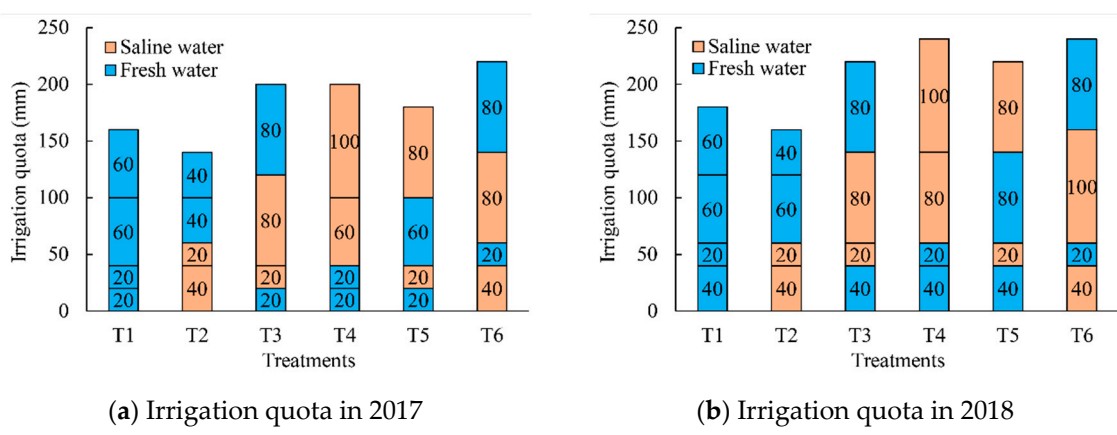

(**a**) Irrigation quota in 2017　　　　　　　　　　　(**b**) Irrigation quota in 2018

**Figure 5.** Irrigation water applied for treatments.

As detailed in Figure 5, the irrigation quota for all treatments at the fruit-set and breaker stages was significantly higher than that at the seeding and flowering stages. The saline water used for the T4 treatment was 160 mm in 2017 and 180 mm in 2018, respectively, which was significantly higher than the other treatments, while the fresh water used for the T4 treatment was the lowest. The irrigation quota for both saline and fresh water at the flowering stage was 20 mm, while at the same growth stage, the irrigation quota for brackish water was no less than that for fresh water. Moreover, the irrigation quota for the T2 treatment was lower than in other alternating fresh-saline water irrigation models; in particular, the irrigation quota for the T2 treatment was lower than that for freshwater irrigation. Additionally, continuous irrigation with saline water at the seeding and flowering stages resulted in a decrease in the irrigation quota at the fruit-set and breaker stages.

### 3.2. Total Nutrients

Soil nutrients were essential for the vegetative and reproductive growth of the plant. The variation of soil nutrients in the main root zone (0–40 cm depth inside the film) at different tomato growth periods was measured and shown in Figure 6.

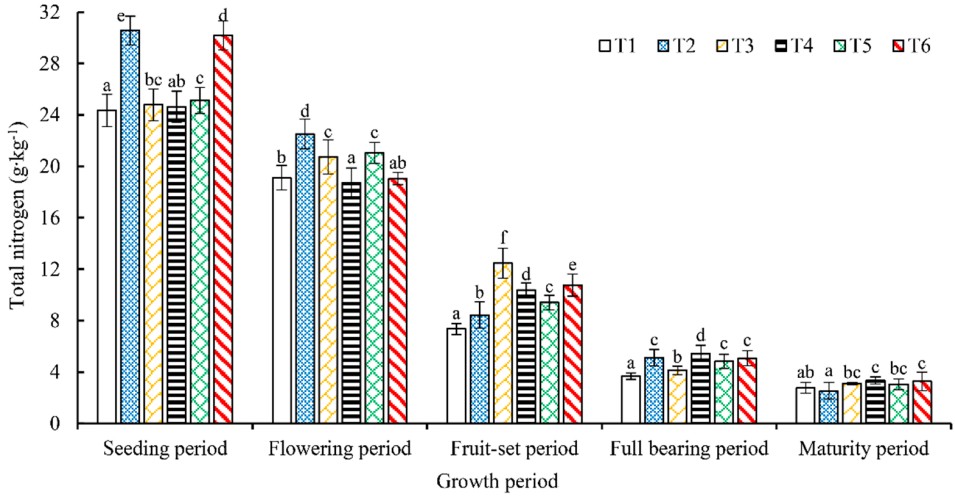

(**a**) Variation of soil total nitrogen in 2017

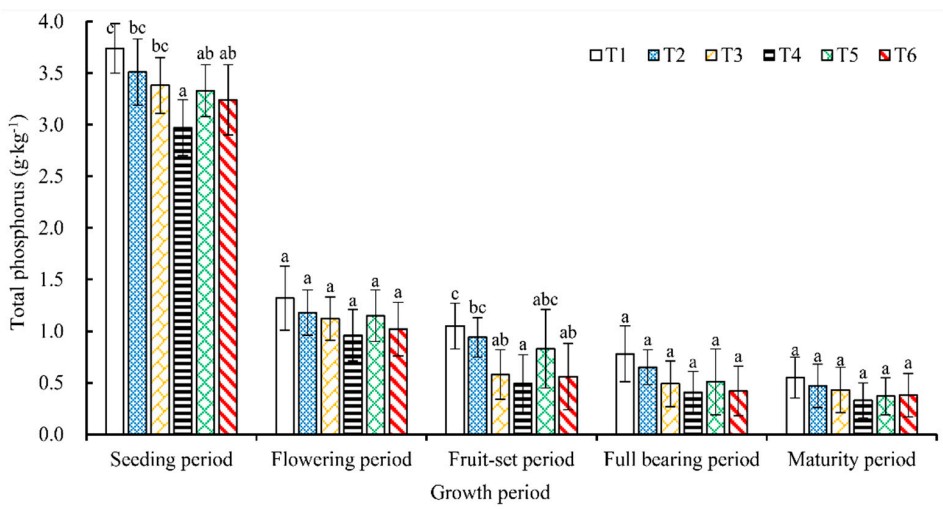

(**b**) Variation of soil total phosphorus in 2017

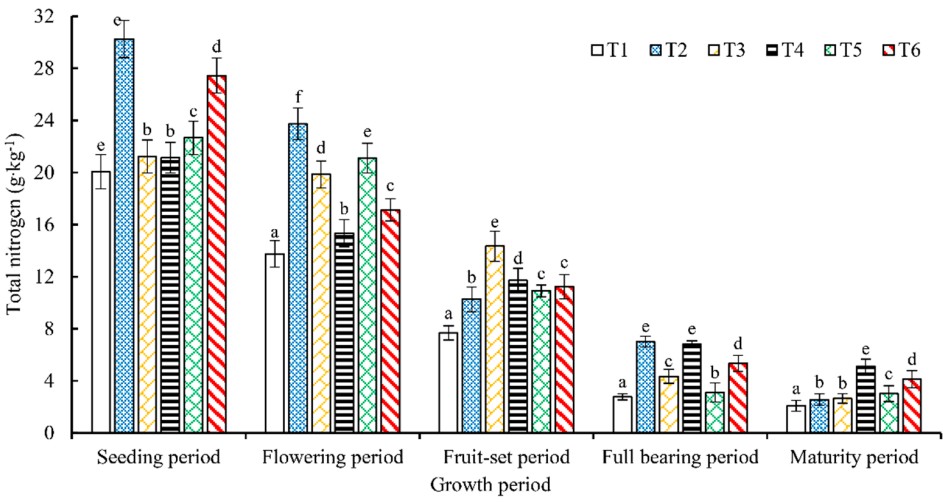

(**c**) Variation of soil total nitrogen in 2018

**Figure 6.** *Cont.*

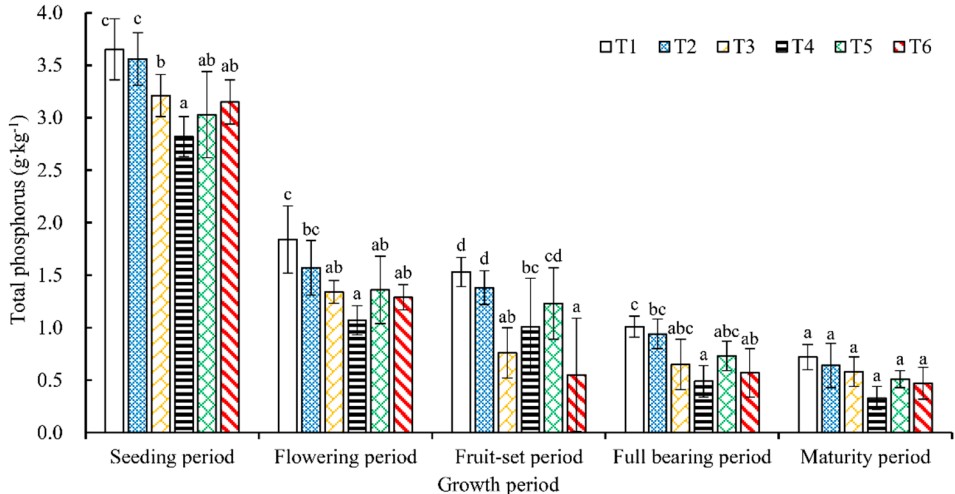

(**d**) Variation of soil total phosphorus in 2018

**Figure 6.** Variations in soil total nutrients. Note: The same letter above the error bars at the same growth stage represents no significant difference at $p \leq 0.05$.

As shown in Figure 6, the total nitrogen content in the root zone decreased gradually along the tomato growth period, and reached a minimum at the maturity stage. Obviously, the value of soil total nitrogen was raised for saline water irrigation from the seedling stage to the fruit-set stage, so the impact of saline water irrigation on soil nitrogen content mainly occurred before the breaker stage.

Similarly, the total phosphorus content in the root zone decreased along the tomato growth period; in particular, the total phosphorus decreased sharply at the seedling period. The soil total phosphate content for the T1 treatment was higher than for the other treatments along the growth period; the content of soil total phosphate for the T2 treatment took second place, while the value of soil total phosphate concentration for the T4 treatment was the lowest.

### 3.3. Salt Ions

In general, there are eight main ions considered in the soil: $HCO_3^-$, $CO_3^{2-}$, $SO_4^{2-}$, $Cl^-$, $Ca^{2+}$, $Mg^{2+}$, $K^+$, and $Na^+$. Since $CO_3^{2-}$ is not stable and the content in the soil usually close to 0, it was not considered in this paper; the other seven main ions' contents in treatments at different growth periods are shown in Figure 7.

As we can see from Figure 7a,c, the content of $Na^+$, $K^+$, $Mg^{2+}$, and $Ca^{2+}$ for treatments in 0-40 cm topsoil increased with growth, and the contents were higher for saline water irrigation than freshwater irrigation at the same growth stage. $Na^+$ had the most obvious soil-soluble cations that were influenced by saline water. The higher the brackish water irrigation quota, the more $Na^+$, $K^+$, $Ca^{2+}$ and $Mg^{2+}$ was brought into the topsoil.

As detailed in Figure 7b,d, the soil $HCO_3^-$ content reached its maximum at the seedling and flowering stages, while the content for treatments during the growth period had little difference. However, the content of soil $Cl^-$ increased gradually over the tomato growth period; in particular, soil $Cl^-$ content for saline water irrigation was higher than that for fresh water. Moreover, the content of $SO_4^{2-}$ for treatments in 40 cm topsoil first decreased and then increased slowly over the growth period of the tomato, and reached a minimum at the breaker stage; the soil $SO_4^-$ tended to increase with the amount of saline water applied at the same stage.

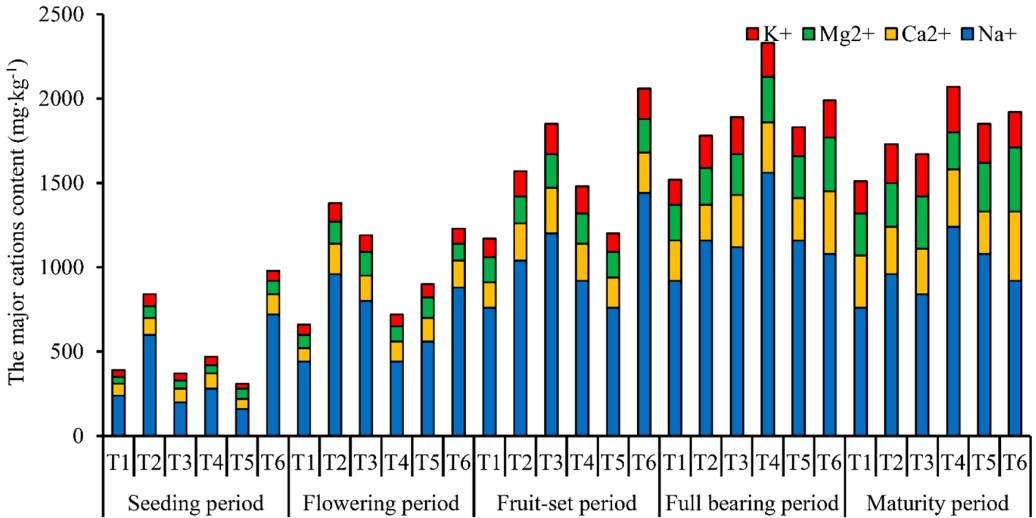

(**a**) The main cations content for treatments in the growth period of 2017

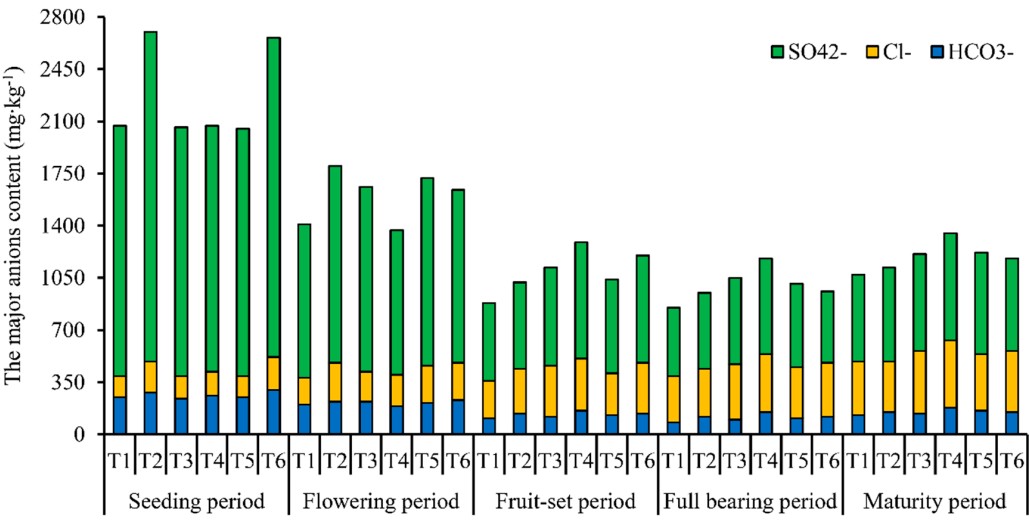

(**b**) The main anions content for treatments in the growth period of 2017

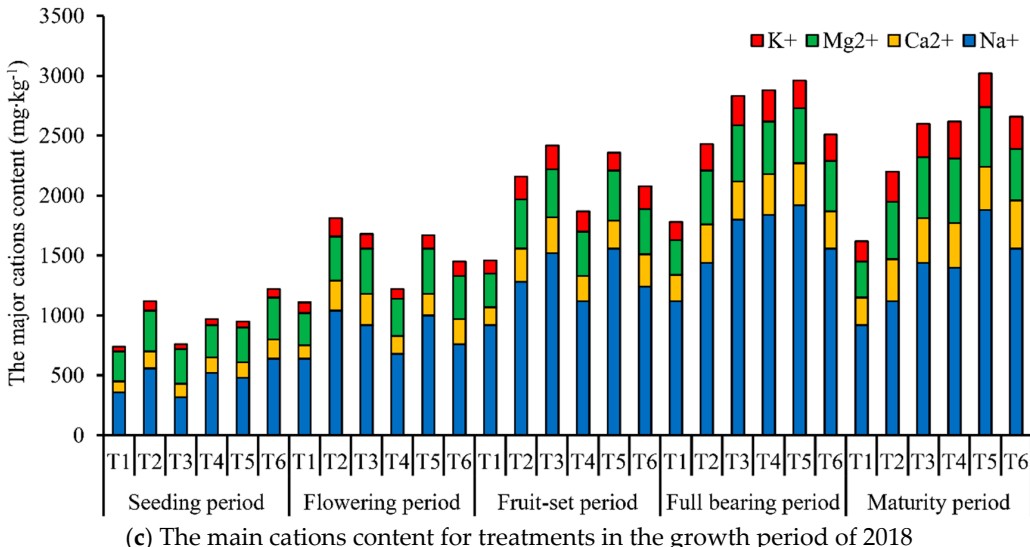

(**c**) The main cations content for treatments in the growth period of 2018

**Figure 7.** *Cont.*

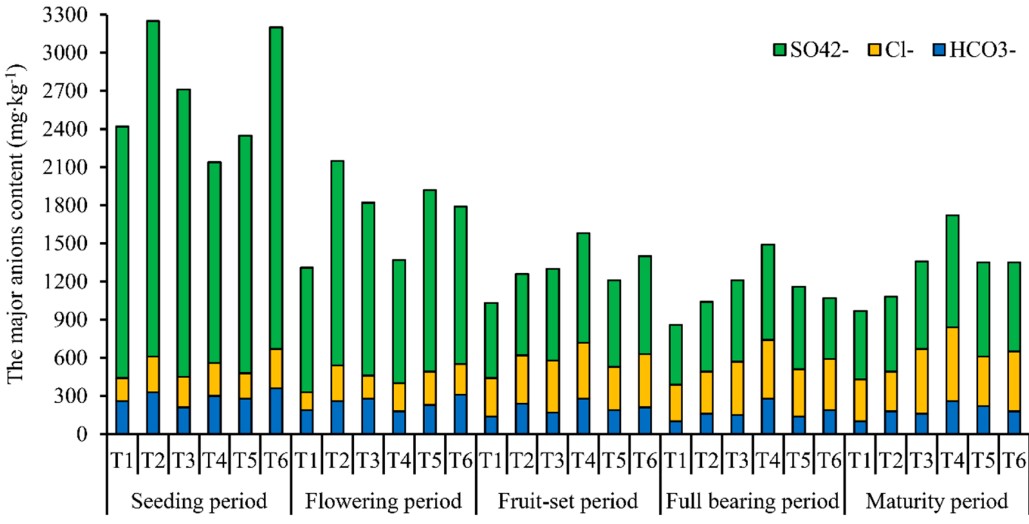

(**d**) The main anions content for treatments in the growth period of 2018

**Figure 7.** Distribution of the main soil ions for treatments.

## 3.4. SAR

To make a further analysis of the effect of alternating irrigation with saline water and fresh water, the soil SAR was measured by soil ions content and shown in Table 6.

**Table 6.** Soil SAR of root zone, unit: $(meq \cdot L^{-1})^{1/2}$.

| Year | Treatment | Seedling Stage | Flowering Stage | Fruit-Set Stage | Breaker Stage | Maturity Stage |
|------|-----------|----------------|-----------------|-----------------|---------------|----------------|
| 2017 | T1 | 1.411 ± 0.34bc | 2.071 ± 0.14a | 2.612 ± 0.11a | 2.604 ± 0.12a | 1.938 ± 0.43a |
| | T2 | 2.802 ± 0.82d | 3.314 ± 0.12c | 3.241 ± 0.32b | 3.321 ± 0.55cd | 2.471 ± 0.26bc |
| | T3 | 1.076 ± 0.09ab | 2.809 ± 0.26b | 3.358 ± 0.34b | 2.890 ± 0.43ab | 2.059 ± 0.48a |
| | T4 | 1.462 ± 0.12c | 1.841 ± 0.22a | 2.774 ± 0.26a | 3.916 ± 0.36e | 2.207 ± 0.15ab |
| | T5 | 0.870 ± 0.14a | 2.088 ± 0.12a | 2.520 ± 0.22a | 3.088 ± 0.24bc | 2.742 ± 0.47cd |
| | T6 | 3.110 ± 0.15d | 3.347 ± 1.02c | 3.134 ± 0.14b | 3.470 ± 0.24d | 2.958 ± 0.32d |
| 2018 | T1 | 1.099 ± 0.02ab | 1.859 ± 0.23a | 2.547 ± 0.06a | 2.903 ± 0.18a | 2.341 ± 0.54a |
| | T2 | 1.448 ± 0.26bc | 2.429 ± 0.56b | 2.835 ± 0.23a | 3.026 ± 0.46a | 2.270 ± 0.36a |
| | T3 | 0.903 ± 0.15a | 2.116 ± 0.47ab | 3.361 ± 0.18b | 3.725 ± 0.52c | 2.834 ± 0.25b |
| | T4 | 1.484 ± 0.26bc | 1.810 ± 0.28a | 2.678 ± 0.24a | 3.861 ± 0.67c | 2.701 ± 0.38b |
| | T5 | 1.332 ± 0.23bc | 2.411 ± 0.12b | 3.517 ± 0.32b | 3.950 ± 0.85c | 3.741 ± 0.44d |
| | T6 | 1.614 ± 0.15c | 1.836 ± 0.26a | 2.836 ± 0.28a | 3.374 ± 0.67b | 3.209 ± 0.08c |

Note: Values in a column in the same year followed by the same letter are not significantly different at $p \leq 0.05$.

As shown in Table 6, the soil SAR of the root zone was lower than 10 $(meq/L)^{1/2}$, even though the soil SAR increased due to brackish water irrigation, and the soil structure for all treatments with brackish water irrigation was hardly affected; moreover, the soil SAR increased gradually and reached a maximum at the breaker stage, then decreased. As the present paper only focused on the soil SAR in the root zone, further studies on the variation of SAR outside the film are required.

## 3.5. Total Salt

To determine the influence of saline water irrigation on soil salinity, the total salt content of the two soil layers (0–40 cm and 40–100 cm) inside and outside the film was measured after harvest; the results are shown in Figure 8.

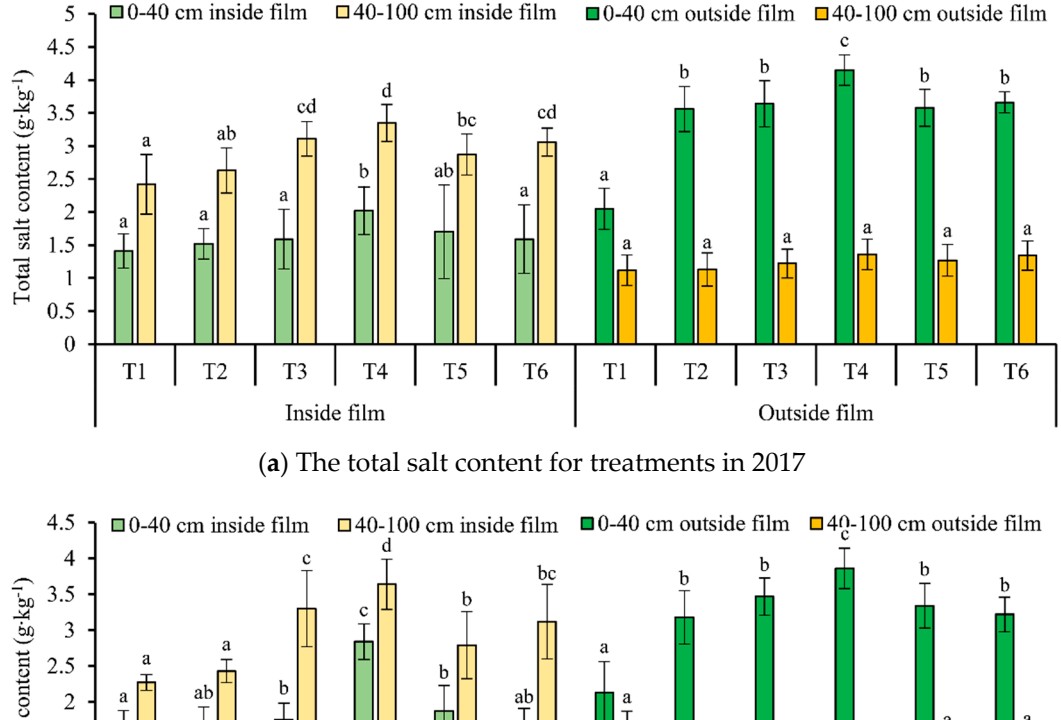

(**a**) The total salt content for treatments in 2017

(**b**) The total salt content for treatments in 2018

**Figure 8.** Total salt content of soil profile for treatments. Note: The same letter above the error bars with the same legend represents no significant difference at $p \leq 0.05$.

After tomato harvesting, the soil salt content of the upper layers both inside and outside the mulch for the T1 and T4 treatments attained the minimum and maximum values, respectively. Meanwhile, the soil salt content at 40–100 cm inside the film for the T1 and T4 treatments achieved the minimum and maximum, respectively; however, soil salt at 40–100 cm outside the mulch showed little difference between treatments.

### 3.6. Tomato Production

Compared with freshwater irrigation, saline water irrigation has dual effects on the plant: on the one hand, more salt was carried into the topsoil, which was not conducive to root water uptake; on the other hand, some salt ions of brackish water are beneficial for the vegetative and reproductive growth of the tomato. The fruit yield of the tomato is listed in Table 7.

**Table 7.** Tomato fruit yield.

| Treatment | 2017 | | 2018 | |
|---|---|---|---|---|
| | Fruit Yield on Each Plant (g/plant) | Tomato Production ($10^4$ kg·ha$^{-1}$) | Fruit Yield on Each Plant (g/plant) | Tomato Production ($10^4$ kg·ha$^{-1}$) |
| T1 | 45 ± 5c | 9.565 ± 0.326c | 43 ± 4c | 9.443 ± 0.423d |
| T2 | 29 ± 3a | 7.821 ± 0.345a | 32 ± 3a | 8.012 ± 0.541a |
| T3 | 36 ± 4b | 8.646 ± 0.432b | 35 ± 2ab | 8.632 ± 0.318c |
| T4 | 40 ± 4b | 9.302 ± 0.517c | 38 ± 2b | 9.264 ± 0.426d |
| T5 | 37 ± 3b | 8.543 ± 0.429b | 36 ± 3ab | 8.412 ± 0.445bc |
| T6 | 37 ± 2b | 8.305 ± 1.338b | 38 ± 5b | 8.226 ± 0.439ab |

Note: Values in a column followed by the same letter are not significantly different at $p \leq 0.05$.

According to Table 7, the tomato production for treatments was in the order of T1 > T4 > T3 > T5 > T6 > T2, and the fruit yield on each plant for the T4 treatment was the highest among alternating irrigation models in both 2017 and 2018. Compared with the T1 treatment, the tomato production for the T4 treatment decreased by just 2.75% in 2017 and 1.90% in 2018, respectively, while the tomato production for the T2 treatment decreased by 18.2% in 2017 and 15.15% in 2018, respectively.

*3.7. Fruit Quality*

To estimate the fruit quality, the common parameters ($T_{SS}$, lycopene, $T_s$, and $T_a$) were measured after tomato harvesting, and the results are shown in Table 8.

**Table 8.** Fruit quality parameters of tomato.

| Treatments | 2017 | | | | 2018 | | | |
|---|---|---|---|---|---|---|---|---|
| | $T_{SS}$ (%) | Lycopene (mg/100 g) | $T_S$ (%) | $T_a$ (%) | $T_{SS}$ (%) | Lycopene (mg/100 g) | $T_S$ (%) | $T_a$ (%) |
| T1 | 7.94 ± 1.23a | 6.50 ± 0.54a | 7.12 ± 0.82a | 1.54 ± 0.06b | 8.04 ± 1.02a | 6.54 ± 0.24a | 7.24 ± 0.32a | 1.47 ± 0.25ab |
| T2 | 8.04 ± 2.45a | 6.92 ± 0.43b | 7.80 ± 0.35c | 1.30 ± 0.04ab | 8.02 ± 1.14a | 7.05 ± 0.26b | 7.76 ± 0.43b | 1.28 ± 0.13b |
| T3 | 8.11 ± 2.34a | 7.12 ± 1.05b | 7.90 ± 0.76cd | 1.23 ± 0.24ab | 8.18 ± 0.36a | 7.08 ± 0.18b | 7.88 ± 0.56bc | 1.31 ± 0.24ab |
| T4 | 8.30 ± 3.04a | 7.70 ± 0.68c | 8.20 ± 1.12d | 0.98 ± 0.17a | 8.34 ± 0.85a | 7.53 ± 0.19c | 8.17 ± 0.78c | 1.04 ± 0.16a |
| T5 | 8.02 ± 2.51a | 7.22 ± 1.24b | 7.94 ± 0.72cd | 1.32 ± 0.34ab | 8.12 ± 0.73a | 7.11 ± 0.12b | 8.02 ± 0.49bc | 1.35 ± 0.34ab |
| T6 | 8.18 ± 2.25a | 7.05 ± 0.57b | 7.46 ± 0.46b | 1.22 ± 0.22ab | 8.20 ± 0.48a | 6.98 ± 0.07b | 7.38 ± 0.67a | 1.18 ± 0.27ab |

Note: Values in a column followed by the same letter are not significantly different at $p \leq 0.05$.

As shown in Table 8, the $T_{SS}$, lycopene, and $T_S$ of tomato fruit for alternating fresh-saline water irrigation was higher than that for freshwater irrigation at all growth stages; moreover, the $T_{SS}$, lycopene, and $T_S$ were highest for the T4 treatment. Additionally, the $T_a$ for alternating fresh-saline water irrigation was lower than that for freshwater irrigation, and the T4 treatment reached the minimum value.

**4. Discussion**

With the same designed threshold (−25 kPa) at each growth stage, the T4 treatment consumed more saline water than other alternating irrigation treatments, and saved more fresh water than other treatments, which was in accord with the large amount of irrigation water consumed at the fruit-set and breaker stages. Moreover, continuous irrigation at the seeding and flowering stages resulted in a reduction of the irrigation quota at the flowering stage. At the same growth stage, the irrigation quota for saline water was no less than that for fresh water, which indicated that saline water irrigation tends to improve the soil matric potential of the root zone more easily than fresh water, especially during growth periods with high water consumption such as the fruit-set and breaker stages.

The value of soil total nitrogen content for saline water irrigation was higher than that of freshwater irrigation at each growth period; conversely, the total phosphate concentration for saline water irrigation was lower than that of fresh water. Additionally, the total nitrogen content in the root zone decreased gradually along with the tomato growth period, and reached a minimum at the maturity stage, which was consistent with the increase in nitrogen necessary for plant tissues and fruit during the growth

stages. The results indicated that saline water irrigation may contribute to a certain increase in soil nitrogen and a certain decrease in soil phosphorus, which was in line with the study of Tian et al. (2018), who has studied the effects of soil nutrients under saline water irrigation in 2016 [42]. The ions of saline water accumulated in the topsoil and then affected the soil nitrogen contents, especially at the seedling and flowering stages. Taking account of the nitrogen in the topsoil, saline water irrigation was recommended before the breaker stage (T2, T3, T6 treatment), because of an increase in the soil nitrogen content. The content of phosphate in the topsoil decreased due to saline water irrigation, which was in line with the study of Tian et al. (2018) [42]. As reported by Jin et al. (2014), there are two possible reasons for the influence of saline water irrigation on soil total phosphate content: on the one hand, the soil ion content tends to increase with the amount of saline water applied, which results in more dissociated phosphorus being adsorbed by soil ions; on the other hand, irrigation with brackish water may decrease the soil phosphorus content by reducing the activity of soil microorganisms such as ammonifying bacteria and the bacteria that decompose organic phosphorus and inorganic phosphorus [43].

Soil $Ca^{2+}$ and $Mg^{2+}$ increased over the growth period, and the content for saline water irrigation was higher than that for freshwater irrigation at the same growth stage, which may arise from the strong adsorption capacity of $Ca^{2+}$ and $Mg^{2+}$ with soil colloids, and the rich content of irrigation water. As soil $Mg^{2+}$ accumulation used to have a deleterious effect on the aggregate soil stability, such as via high swelling by expanding clays, extra measures such as soil amendment, supplementary irrigation, etc., should be taken to decrease the $Mg^{2+}$ content of the topsoil. Soil $Na^+$ and $K^+$ increased over the growth period of the tomato, and the content of topsoil $Na^+$ and $K^+$ increased especially for the saline water irrigation, which was in accordance with the study of Karlberg and de Vries (2004), who reported that with saline water irrigation, sodium ions ($Na^+$) tend to accumulate and concentrate in the topsoil due to the soil evaporation [44]. With the characteristics of low ionic charge, small hydration, and large radius, soil $K^+$ and $Na^+$ usually have a weak adsorption capacity with soil colloids, [45,46]; accompanied by high background values of $K^+$ and $Na^+$ in the topsoil and the high concentration of $K^+$ and $Na^+$ in the saline water, $K^+$ and $Na^+$ tend to accumulate in the topsoil. The accumulation of $Na^+$ usually results in dispersion and expansion of soil particles, such as plugging of pores owing to clay particles [45], which may result in the destruction of soil aggregates, a reduction in soil hydraulic conductivity, and a decrease in available water capacity [47–49]; moreover, the accumulation of $Na^+$ may lead to negative effects on the soil chemical characteristics, such as a reduction of available nitrogen (N), phosphorus (P), and potassium (K) [50,51].

Soil $HCO_3^-$ decreased over the growth period, which indicated that the topsoil gradually transitions from alkaline to neutral in the course of the cultivation. The content of $Cl^-$ increased over the tomato growth period, which may be ascribed to the weak adsorption capacity of $Cl^-$ to soil colloids, and the high level of $Cl^-$ in saline water. The content of $SO_4^{2-}$ first decreased and then increased, which may be a result of the base fertilizer of $K_2SO_4$ and the higher content of $SO_4^{2-}$ in saline water than in fresh water.

Plant tissues usually accumulate $Na^+$ and $Cl^-$, which are induced by saline water, which will cause osmotic stress, ion toxicity, and ion imbalance in tissues [52]; the uptake of the nutrients can be interrupted by competitive membrane selectivity or interaction [51]. Moreover, $Na^+$ toxicity can lead to $K^+$ deficiency, which will result in water loss and necrosis [53]. Additionally, the stomatal conductance and photosynthesis rate of plants may decrease due to the decline in carbon dioxide ($CO_2$) availability that results from $Na^+$ toxicity [54]. Moreover, the photochemical system or metabolism may be interfered with [51]. Thus, saline water irrigation at the seedling stage of tomatoes is not suggested.

The soil SAR of the root zone was less than 10 $(meq/L)^{1/2}$, which indicated that an irrigation schedule alternating brackish water and fresh water will not result in soil alkalization in the root zone; however, the application of saline water will cause an increase in the SAR value in the root zone at each growth stage, which is in line with the high content of salt ions contained in brackish water. To avoid topsoil alkalization, saline water irrigation at the seedling and breaker stages was

not suggested; however, there is supplementary irrigation after the harvest every year with a large amount of water (90 mm) for leaching the soil salt, so saline water irrigation with a proper schedule is practicable. However, close attention should be paid to the damage soil sodium salt may do to the soil environment. After the tomato harvest, the total content of soil salt at 40–100 cm inside the film and surface layer outside the mulch appeared to be relatively high, and the total salt content increased with the irrigation quota of brackish water; moreover, there was little difference from the soil salt content at 40–100 cm outside the film, which arises from leaching and evaporation. To reduce the accumulated soil salt, an alternating irrigation schedule of the T2 treatment is recommended. Compared with fresh water, there are more soluble salts in saline water, so saline water irrigation not only dilutes the soil solution by increasing the topsoil moisture, but also carries dissolved salt into the topsoil inside the mulch. Additionally, soil salt in the root zone migrates into the deeper layers inside the film and the soil layers outside the film. Moreover, the higher the irrigation quota, the more sufficient the leaching, and the soil salt tends to migrate to the margin of the wetting body, which results in a higher content of soil salt in the upper layers outside the film. Generally, drip irrigation usually supplies water to the topsoil frequently, so soil salt is sufficiently leached; moreover, some salt moves to the fringes of the root zone along with soil water, so the desired desalination circumstance was formed near the emitter for better plant growth [55]. It is evident from the results that the more saline water is applied, the more soil salt accumulates; compared with freshwater irrigation, alternating saline water irrigation tends to promote soil salt accumulation, which corresponds to a higher dissolved salt content in brackish water. To reduce the accumulated soil salt, the alternating irrigation schedule of the T2 treatment is recommended.

Additionally, compared with freshwater irrigation, the tomato fruit yield for alternating fresh-saline water irrigation models decreased by 2.75–18.20% in 2017 and 1.90–15.15% in 2018, respectively, which indicated that alternating fresh-saline water irrigation will result in reduced tomato production; however, with proper irrigation models, this reduction can be controlled to a certain extent. The $T_{SS}$, lycopene, and $T_S$ of tomato fruit for the T2, T3, T4, T5, and T6 treatments were higher than for the T1 treatment, while the $T_a$ for freshwater irrigation was higher than for alternating fresh-saline water irrigation, which indicated that trace elements of brackish water are beneficial in terms of the increase in $T_{SS}$, lycopene, and $T_S$, a result identical to the study of Abdel Gawad, who reported that the soluble solid content of tomato fruits rose when exposed to a certain amount of brackish water [56].

## 5. Conclusions

The alternating fresh-saline water irrigation model for the T4 treatment (fresh water in the seedling and flowering stages, saline water in the fruit-set stage breaker stages) was more freshwater-saving than other treatments, and was more suitable for high tomato production and improvement of fruit quality. According to the effects of alternating irrigation models on topsoil nutrient and the SAR of the topsoil, the alternating irrigation model for the T3, T4, and T5 treatments was better than other treatments. Ultimately, the alternating fresh-saline water model of the T4 treatment was suggested to be the proper irrigation style for the utilization of shallow ground saline water in HID.

**Author Contributions:** Conceptualization, J.L. and Z.Q.; methodology, J.C.; software, J.L.; validation, J.L., J.C. and P.H.; formal analysis, J.L.; investigation, P.H.; resources, J.L.; data curation, Z.Q.; writing—original draft preparation, J.L.; writing—review and editing, Z.Q.; visualization, N.Z.; supervision, J.L.; project administration, J.C. and S.W.; funding acquisition, J.C. and Z.Q.

**Funding:** This research was funded by "Major Project in Key Research and Development Program of the Ningxia Hui Autonomous Region," grant number "2018BBF02022"; "The National Key Research and Development Program of China," grant number "2016YFC0501301"; "the Fundamental Research Funds for the Central Universities," grant number "2019B70314"; "The National Science Foundation Project," grant number "41761050" and "Postgraduate Research & Practice Innovation Program of Jiangsu Province," grant number "SJKY19_0484".

**Acknowledgments:** We are especially grateful to Bin Du for her constructive comments during the review process. We also thank Yongping Huang for his help in the field.

**Conflicts of Interest:** The authors declare no conflict of interest.

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
