# Peer review of "Effects of Alternating Irrigation with Fresh and Saline Water on the Soil Salt, Soil Nutrients, and Yield of Tomatoes"

_water, doi:10.3390/w11081693_

Round 1
Reviewer 1 Report
The manuscript deals with the investigation of the effects of alternating irrigation (waters with different quality) during different growth stages on several agronomical parameters of a widely cultivated crop (tomato plants) and falls within the scope of Water journal.
The manuscript could be considered as an original contribution since it presents interesting results on the effects of alternating fresh and saline water irrigation on soil, considering different development stages on tomato plants.
Although the research is interesting, because it offers some interesting and new information that it could worth publishing and can be very useful in arid and semi-arid regions, the manuscript is not very well written, the results are limited and conclusions are not supported by data and the paper has important weakness that I detail below. Moreover, this paper requires a better interpretation of the results according to statistical analysis.
For these reasons, mainly I recommend that the paper be rejected
Apart from the above, other specific items should be taken into consideration
Specific amendments to correct errors of fact:
Title
They should add the word “tomato”, as the experiment is carried out only in this crop.
Abstract
Summary is clear and satisfactory. It is in general sufficiently informative. However, they should add in which crop are the experiments carried out (tomato). They should also add the electrical conductivty (EC) or TDS (g/l) for each kind of water (fresh and saline water irrigation).
–(line 30) in the abstract, They should explain what is “designed alternating irrigation models”
-line 31. Are authors refering to T4 in this sentence? They have to give the reasons because t4 is suggested like the best irrigation model in tomato cultivation in HID. In which parameters is t4 better than the other treatments: quality, quantity, less or more content of some nutriente/ion? Add this information.
Introduction
-The introduction provides an adequate background about the problem with shortage of fresh water and the effect of salinity in soil. Although a revision of English language and spell check are required: (. However, line 39; garricultural, line 48; applys large numbers of water, line 61; seeding, line25; model model , line 397…)
- line 90.this sentence about transplantation has no sense. The experiment does not study the effect of transplantation in alleviating the stress of plants.
- line 91, revise this sentence : alleviate the stress resistance of tomato to soil salt. It has no sense. Alleviate the salt stress or increase the resistance.
- at the end of this section, they have to add that the goals of this study are for tomato cultivation, not for all crops.
-line 104, add “ in soil”.
- line 106, in my opinion “ different alternating irrigation models” is not very appropriate. In this context would be better “ with different water resources” or with waters with different quality, water sources.
Material and methods
They have to add some details about irrigation
Line 126. Add the electrical conductivity of the each type of water.
Table 4. Revise line 3 ( K or na). it must be a mistake. In the column names, what is the meaning of “project”?.
Line 156-169. Is this paragraph about the traditional tomato cultivation in this area or is it the description of the conditions in the experiment? It has to be clearer.
In the traditional cultivation the irrigation is surface flood irrigation (900+01200 m3 /ha), while your experiment is with drip irrigation ( between 1300 and 2100 m3/ha)????
What the frequency of irrigation was? Every day , every week? Add some information about this.
Line 178. When the grafting was?
-how many tensiometers in each treatment? Did all treatments start the irrigation at the same time (when the mean of all tensiometer were lower than -25 kpa) or each treatment start when its tensiometers were lower than -25 KPa?. It has to be clearer.
Did all treatments have the same date for the beginning and ending of each growth period ?
Meteorological data, values for tensiometers and soil basic properties are not “measurements”, they are “experimental conditions”. Values for tensiometer are used for scheduling irrigation and meteorological and soil properties are described in table 1, 2 and 3, and figure 2 as description of experimental area ( line 117, line 119). If the authors show these data in this way (description of the experimental area), these data can not be shown again as results or measurement during experiment. In this manuscript, the only measurements made during the experiment are soil salinity (2.4.4.) and soil nutrient content ( 2.4.5). results are a bit limited, they should have added some measurements in plant. Figures from 1-4 and tables from 1-5 are data for the description of the experimental area and experimental conditions. Results are fig 4 to8 and table 6.
Results
authors have to add error bars and/or letters in the figures and error standards or letters in tables. In the current form it is impossible to know if there are significant differences or not between treatments.
Figure 5 has to improve, legend does not see properly. Moreover, figure 5 has a lot of useful information that can be discussed deeper.
-line 23-242. This sentence should be placed in discussion section, it is not a result. The same in line 243-244. The structure of the manuscript is not appropriate.
. In conclusion they write that “ t4 is fresh-water saving”. They should be written in result section when authors describe figure 5.
- line 242-245. This sentence is not logic. Rewrite. They write “the irrigation for t2 was the lowest, which suggest that tis not suitable…”. It is not coherent. Maybe they wanted to say “ t2 is not suitable because the survival rate may be reduced…”. In any case, it is also a mistake, because the survival rate is not measured in this experiment, or at least, it is not shown in this manuscript.
- line 251-252. This sentence is not a result. This kind of sentences should be in discussion section.
Line 253, “ soil N content for T1 was lower than other treatments”. According to figure 6, it is false. I can not see the differences between treatments. Be careful with the statistical analysis. – line 258, ¿nitrogen or phosphorus content?
- line 288, “soil structure….” . line 290-291. It is for discussion section. In which treatment soil structure was hardly affected? In all treatments with brackish water irrigation? T2 to t6?, it has to be clear.
Line 296. If some measurement are made inside and outside the fi, they should have been explained in material and methods.
Line 303-305. Discussion instead results.
Discussion.
Line 308. Revise this sentence · “fluctuate more frequently” has no sense.
-line 311-313. The manuscript does not show data about survival rate. The discussion and conclusions about the most suitable treatment cannot be based on this parameter.
Line 314, 318. This sentence is not supported by data presented in figure 6. According to fig 6 I can not say that “saline water irrigation may contribute to a certain increase of soil nitrogen and a certain decrease of soil phosphorus”. Authors have to add letters and describe results according with statistical analysis.
Line 321, what “ they made impacts on soil..” means?
Line 322. “Taking account of the nitrogen in the top soil, saline water irrigation was recommended before full bearing stage of tomato”. They have to give reason ( because an increase / decrease of …). What treatments are authors referring: all treatments except T4 and T5? It has to be clearer.
According to figure 6 (N), I do no see differences between t4-t5 and the rest of saline treaments. This sentence is not true according to data presented in figure 6a and 6c.
Is Nitrogen in the top soil the same as soil total nitrogen, data in figure 6?
-line 323-324. “The content of phosphate in topsoil decreased due to saline water”. For this statement, we would need letters, I am not sure that all treatments (t2-t6) and in all stages were lower than t1.
- line 330-332.. this sentence is not supported by the data.
- line 351-352. Authors have to add letters.
-respect to the poison function. Poison is not an appropriate word. Toxic?
“brackish water irrigation at tomato seeding stage is not recommended”. Are they referring to T2 and T6?. Clear. In figure 7, I can not see if the Na content in t2 and t6 is lower or higher than the other treatments!!!.
Conclusion
It should be rewrite
If t4 is fresh water saving, it has to be written in results section. Seed viability is not shown, for this, it can not be the reason for select this treatment like the best irrigation model. Authors do not shown any data about plant behaviour, only for soil.
Line 398-399, “ t2 is better for alleviation”. Better than the rest of treatments? The best? For which reason?.
- line 396,” damage of na and cl to plant tissue” !!! they is not shown in this manuscript
Line 394- 369: “Considering …the SAR of the topsoil, the alternating irrigation model model for T3, T4 and T5 treatment was better than other treatments”. In table 2017, I cannot see that.
Author Response
Dear Reviewer,
Thank you very much for your careful review of our manuscript, we appreciate you very much for the suggestions.
I have revised the manuscript carefully according your suggestions during the past seven days, and hope you can give us a chance.
The main corrections in the manuscript are as flowing:
1. Line 3, the expression “Yield of Tomato” was added into the Title.
2. Line 17-18, the sentence “To study the effect of alternating irrigation models on soil nutrients and soil salts,…” was replaced by “To study the effect of alternating irrigation models on soil nutrients, soil salts and yield of tomato with fresh water (total dissolved solids of 0.50 g·L-1) and saline water (total dissolved solids of 3.01 g·L-1),…”, which contains the crop species and irrigation water salinity.
3. Line 30, the word “experimental” was added before “designed alternating irrigation models”, which show clearly that the “designed alternating irrigation models” was the six drip irrigation models as mentioned above.
4. Yes, the expression “drip irrigation with fresh water at seeding to flowering stage and saline water at fruit-set to breaker stage” corresponding to the T4 treatment.
Line 31, the analysis of tomato yield and fruit quality was added in the manuscript. Though the tomato yield for fresh-saline water alternating irrigation treatments decreased, compared with fresh water irrigation, T4 treatment just decreased by 2.75% in 2017 and 1.9% in 2018. Additionally, the total soluble solids, lycopene and total sugar of tomato fruit for T4 treatment was higher than other treatments.
5. The phase “seedling stage” has been replaced by “seedling stage” in the whole document.
6. Line 39, the unsuitable expression “…HID (Hetao Irrigation District), however, with…” has been corrected to “…HID (Hetao Irrigation District). However, with…” in the Introduction part.
7. Line 48, the misspelling phase “garicultural irrigation” has been corrected to “agricultural irrigation” in the Introduction part.
8. Line 61, the misspelling word “applys” in line 48 has been corrected to “applies” in the Introduction part.
9. Line 99, the misspelling word “maximun” in line 99 has been corrected to “maximum” in the Introduction part.
10. Line 101, the redundant word “and” has been delated in the Introduction part.
11. Line 245, the misspelling word “damge” has been corrected to “damage” in the Results part.
12. Line 279, the misspelling word “minimun” has been corrected to “minimum” in the Results part.
13. Line 397, the redundant word “model” has been delated in the Conclusion part.
14. Line 412, the redundant word “to” has been delated in the Acknowledgments part.
15. Line 428, the misspelling word “cron” has been corrected to “corn” in the Reference part.
16. Line 541, the misspelling word “Ressource” has been corrected to “Resource” in the Reference part.
17. Line 90-91, the sentence “studies have shown that, transplantation can be adopted to alleviate the stress resistance of tomato to soil salt [26],” from line 90 to line 91 has been delated, as it has no sense to the study.
18. Line 103, the expression “for tomato cultivation” was added after “…based on growth stages under mulched drip irrigation”.
19. Line 104, the expression “in soil” was added after “…on total nutrient and available nutrient” in line 104.
20. Line 106, the unsuitable expression “…with different alternating irrigation models.” was replaced by “…with different water resources.” in line 106.
21. Fresh water with an amount of 900 m3·ha-1 was applied for supplementary irrigation before the transplanting to leaching the topsoil salt. Then, after the tomato seedlings grafting (May 13, 2017 and 2018), irrigation was triggered by the tensiometers which buried 0.2 m underneath the emitters, as soon as the value of all tensiometers in each treatment were lower than -25 kPa, irrigation was implemented for each treatment with corresponding irrigation water resources (Table 5), and the same irrigation amount was 200 m3∙ha-1. Additionally, none irrigation water was applied for tomato at the ripening stage to avoid rotting.
Table 5. Treatments of saline water on tomato
Treatment | Seedling stage | Flowering stage | Fruit-set stage | Breaker stage | Maturity stage |
T1 | FW | FW | FW | FW | |
T2 | SW | SW | FW | FW | |
T3 | FW | SW | SW | FW | |
T4 | FW | FW | SW | SW | |
T5 | FW | SW | FW | SW | |
T6 | SW | FW | SW | FW |
Note: “FW” is irrigation with fresh water (Yellow River Water, TDS of 0.5 g·L-1), while “SW” is irrigation with prepared saline water (shallow ground saline water, TDS of 3.0 g·L-1)
22. Line 126, the additional expression “(electrical conductivity of 0.79 dS·m-1)” has been added behind “…0.505 g·L-1” in Line 126, meanwhile, the additional expression “(electrical conductivity of 4.70 dS·m-1)” has been added behind “…3.006 g·L-1” in Line 126.
23. Line 131, the unsuitable expression “K++Na+” has been corrected to “K+ or Na+” in line 3 of Table 4; the unsuitable word “Project” has been corrected to “Characteristics” in the column names of Table 4.
24. Line 156, the expression “According to the extensive investigation in 2016,” was added before “The traditional tomato cultivation is generally planted by film covering,…” in Line 156.
25. Line 163, the expression “According to the traditional fertilization schedule, diammonium phosphate…” was replaced by “The fertilization schedule of present field experiment was based on the traditional fertilization schedule. Diammonium phosphate…” in Line 163.
26. Line 169, the unsuitable word “is” has been corrected to “are” in Line 169.
27. Line 157, the total irrigation quota for traditional surface flood irrigation during tomato growth period in HID is 3600 m3·ha-1, not 1200 m3·ha-1. As there are 3 times irrigation during tomato growth period, and the irrigation quota for each application is 1200 m3·ha-1, I have omitted the irrigation times.
28. Line 178-181, the mulched drip irrigation was triggered by “the tensiometers which buried 0.2 m underneath the emitters, as soon as the value of tensiometers for each treatment lower than -25 kPa, irrigation was implemented for each treatment with the corresponding irrigation water resources, and the same irrigation amount was 200 m3∙ha-1”, which has no fixed frequency, as the soil matric potential (SMP) varies with the growth period of tomato, irrigation water salinity, the climate et al.
29. Line 178, the additional expression “(May 13, 2017 and 2018)” has been added behind “…after the tomato seedlings grafting” in Line 178.
30. Line 150, the expression “each treatment was equipped with three vacuum gauge tensiometers which installed at 20 cm underneath the emitter for SMP monitoring” was added at the end of “2.2 Experimental design” in Line 150.
31. Line 179-180, the expression “…, as soon as the value of tensiometers lower than -25 kPa, irrigation was implemented with corresponding irrigation water resources,…” was replaced by “…, as soon as the value of all tensiometers in each treatment were lower than -25 kPa, irrigation was implemented for each treatment with corresponding irrigation water resources,…” in Line 179 and 180.
32. Line 140-142, there was some difference about the date for the beginning and ending of each growth period for all treatments, thus, the average beginning and ending date of each growth period was delated.
33. “2. Materials and Methods” part: “2.1 Experimental area” was renamed by “2.1 Experimental conditions”; “2.4 Observation and equipment” was renamed by “2.4 Measurements”; “2.4.1 Meteorological observation”, “2.4.2. SMP measurement” and “2.4.3. Soil basic properties” were delated, the original part of “2.4.1. Meteorological observation” and “2.4.3. Soil basic properties” were incorporated into “2.1 Experimental conditions”, while the former part of “2.4.2. SMP measurement” was incorporated into “2.2 Experimental design”.
34. The original “Figure 2” is placed in front of “Figure 1”.
35. The measurement of tomato yield was added in “2.4.3. Yield of tomato”,
36. Statistical error calculation and significance analysis has been added in Table 6, Table 7 and Table 8.
37. Figure 5(a) and Figure 5(b) has been updated, and the value has been added in Figure 5(a) and Figure 5(b).
38. Statistical error bars and letters for significance analysis has been added in Figure 6(a), Figure 6(b), Figure 6(c), Figure 6(d), Figure 8(a) and Figure 8(b).
39. Line 236-245, the result for Figure 5 was updated. “As detailed in Figure 5, the irrigation quota for all treatments at fruit-set stage and breaker stage was significantly higher than that at seeding stage and flowering stage. The consumed saline water for T4 treatment was 160mm in 2017 and 180mm in 2018, which was significantly higher than other treatments, however, the consumed fresh water for T4 treatment was the lowest. Both the irrigation quota for saline water and fresh water at flowering stage was 20mm, while at the same growth stage of tomato, the irrigation quota for brackish water was no less than that for fresh water. Moreover, the irrigation quota for T2 treatment was lower than other fresh-saline water alternating irrigation models, especially, the irrigation quota for T2 treatment was lower than that for fresh water irrigation. Additionally, continuous irrigation with saline water at seeding stage and flowering stage resulted in the decrease of irrigation quota at fruit-set stage and breaker stage.”
40. Line 239-242, the sentence “The result indicated that, saline water irrigation tends to improve the soil matric potential of root zone more easily than fresh water, especially at growth period with high water consumption such as fruit-set and breaker stage” in Line 239-242 was moved into the Discussion part.
41. Line 243-245, the sentence “which suggest that it was not suitable for continuous irrigation with saline water at seedling and flowering stage of tomato, the survival rate may be reduced and the tissues of plant may suffer from damage” was delated as no coherent.
42. Line 242, the expression “T4 treatment was more fresh water saving than other treatments” was added after the word “Moreover” in Line 242.
43. Line 251-252, the expression “which was consistent with the increasing of necessary nitrogen for plant tissues and fruit along the growth period” in Line 251-252 was moved into the Discussion part.
44. Line 253, the false sentence about the statistical analysis has been delated.
45. Line 258 the sentence was focused on the total phosphorus, the misspelled word “total nitrogen” has been replaced by “total phosphorus”.
46. Line 288 and Line 290-291 were moved to the Discussion part.
47. Line 288, the original expression “…,and the soil structure was hardly affected,…” was replaced by “…,and the soil structure for all treatments with brackish water irrigation was hardly affected,…”.
48. Soil samples were taken during each growth period at depth of 0-40 cm and 40-100 cm in the two locations which 0 cm (inside the film) and 75 cm (outside the film) from the drip tube for the measurement of soil salinity.
49. Line 303-305, the sentence from Line 303-305 has been moved into the Discussion.
50. Line 307-309, the expression “With the same designed threshold (-25 kPa) at each growth stage, the soil matric potential of the main root zone fluctuate more frequently for saline water irrigation than that for fresh water irrigation at the same growth period, which result in the quota of saline water usually higher than the amount of fresh water applied,…” was revised to “With the same designed threshold (-25 kPa) at each growth stage, the quota of saline water usually higher than the amount of fresh water applied,…”
51. Line 311-313, the sentence “Moreover, to avoid the damage of tomato plant tissues and keep considerable survival rate, continuous irrigation with saline water at seedling and flowering stage of tomato was not suitable” was delated.
52. Line 314-315, the unsuitable expression “Though the total nitrogen and the total phosphorus content in the root zone decreased gradually along with the tomato growth period, while…” was delated.
Statistical error bars and letters for significance analysis has been added in Figure 6(a) and Figure 6(c).
53. Line 321, the unsuitable expression “…then made impacts on…” was replaced by “… then affected the…” in Line 321.
54. Line 322-323, the sentence “Taking account of the nitrogen in the top soil, saline water irrigation was recommended before full bearing stage of tomato” was replaced by “Taking account of the nitrogen in the top soil, saline water irrigation was recommended before breaker stage of tomato (T2, T3, T6 treatment), because an increase of soil nitrogen content”.
55. There were differences between T4 and T5 treatment and the rest of saline treatments, though they were not significant.
56. Yes, the nitrogen in the top soil is the same as soil total nitrogen.
57. According to Figure 6(b) and Figure 6(d), the content of phosphate in topsoil for all treatments were lower than T1 in all growth stages of tomato.
Statistical error bars and letters for significance analysis has been added in Figure 6(b) and Figure 6(d).
58. Line 330-332, the sentence “Considering the soil phosphorus of the top soil for tomato plant, brackish water irrigation is not suggested at seeding stage of tomato” was delated.
59. Statistical error bars and letters for significance analysis has been added in Figure 6(a), Figure 6(b), Figure 6(c) and Figure 6(d)
60. Line 351, the unsuitable word “poison” was replaced by “toxic” in Line 351.
61. Line 352-354, the unsuitable sentence “With respect to the toxic function of Na+ to tomato roots, brackish water irrigation at tomato seedling stage (T2 and T6 treatment) is not recommended” was delated.
62. The Conclusion has been rewritten.
63. Line 242, the sentence “T4 treatment was more fresh water saving than other treatments” was added in Line 242.
64. Line 395, the expression “and beneficial for seed viability of tomato” was delated in Line 395.
65. Line 399, the expression “than other treatments” was added in Line 399, behind “…soil salt accumulation”. The soil salt content for T2 treatment in 0-2500px was lower than other alternating irrigation treatments (Figure 8).
66. Line 398, the expression “than T2, T3, T4 and T5 treatments” was added in Line 398.
67. Line 396, the unsuitable expression “, the damage of Cl- and Na+ to plant tissues,” in Line 396 was delated.
68. Line 395-397, the unsuitable expression “Considering …the alternating irrigation model for T3, T4 and T5 treatment was better than other treatments” has been replaced by “Considering …the alternating irrigation model for T2, T2 and T4 treatment was better than other treatments”.
69. The unsuitable expression “full bearing stage” has been replaced by “breaker stage” in the whole document.
70. Line 51, the original expression “…TDS (Total Dissolved Solids)…” has been replaced by “…total dissolved solids (TDS)…” in the Introduction part.
71. Line 105, the original expression “…SAR (Sodium Adsorption Ratio);” has been replaced by “…sodium adsorption ratio (SAR);” in the Introduction part.
72. Line 114, “Ding, Y.H.; Gao, X.Y.; Qu, Z.Y.; Jia, Y.L.; Hu, M.; Li, C.J. Effects of biochar application and irrigation methods on soil temperature in farmland. Water 2019, 11,449. doi: 10.3390/w11030499” was added as the citation of evaporation in Line114.
73. Line 174, the original expression “… the SMP (soil matric potential)…” has been replaced by “…the soil matric potential (SMP)…” in the Materials and Methods part.
74. The misspelling word “damge” has been corrected to “damage” in the whole document.
75. The analysis of tomato production and fruit quality (total soluble solids (TSS), lycopene, total sugar (Ts) and total acid (Ta)) were added in the manuscript.
76. The reference “Ding, Y.H.; Gao, X.Y.; Qu, Z.Y.; Jia, Y.L.; Hu, M.; Li, C.J. Effects of biochar application and irrigation methods on soil temperature in farmland. Water 2019, 11,449. doi: 10.3390/w11030499” has been added to the Reference part.
77. The reference “Brandt, S.; Pék, Z.; Barna, É.; Lugasi, A.; Helyes, L. Lycopene content and colour of ripening tomatoes as affected by environmental conditions. J. Sci. Food Agric. 2006, 86(4), 568-572. doi:10.1002/jsfa.2390” has been added to the Reference part.
78. The reference “Abdel Gawad, G.; Arslan, A.; Gaihbe, A.; Kadouri, F. The effects of saline irrigation water management and salt tolerant tomato varieties on sustainable production of tomato in Syria (1999-2002). Agricultural Water Management 2005, 78(1-2), 39-53. doi: 10.1016/j.agwat.2005.04.024” has been added to the Reference part.
79. Line 428, the misspelling word “cron” has been corrected to “corn” in the Reference part.
80. Line 472, the misspelling word “elaeagnus” has been corrected to “Elaeagnus” in the Reference part.
81. Line 480, the “doi: 10.1016/j.agwat.2005.04.016” has been added to the twenty-third reference in the Reference part.
82. Line 541, the misspelling word “Ressource” has been corrected to “Resource” in the Reference part.
Once again, thank you very much for your suggestions.
Best regards,
Sincerely,

Reviewer 2 Report
General comments. The manuscript "Effects of alternating irrigation on soil and soil with fresh and saline water" discusses the effect of alternating fresh salt irrigation patterns on soil nutrients and soil salts. An interesting two-year field experiment was conducted in the Hetao irrigation district. The results found that irrigation with salt water tends to have a positive effect on total soil nitrogen and a negative influence on total soil phosphorus at every stage of tomato growth.
The topic is interesting. The proposed methodology is of good technical quality. Overall the paper is well structured. Specific comments are provided in what follows.
Title and abstract.
(i) The paper’s title is suitable, concise and appealing.
(ii) The abstract conveys the purpose of the study in a readable way.
Introduction
· The introduction informs the reader about the objectives of the paper that are well-focused.
Material and methods.
· The analysis is very accurate and precise.
Results and Discussion.
· The discussion of results is very clear,
· The study proposes a very interesting methodology.
· Captions of Figures and Table are adequate.
Conclusions.
· The conclusions are appropriate and focused results.
Bibliography/References are appropriate.
Author Response
Dear Reviewer,
Thank you very much for your careful review of our manuscript, we appreciate you very much for your advice.
I have revised the manuscript carefully during the past seven days, and the main corrections in the manuscript are as flowing:
1. Line 3, the expression “Yield of Tomato” was added into the Title.
2. Line 17-18, the sentence “To study the effect of alternating irrigation models on soil nutrients and soil salts,…” was replaced by “To study the effect of alternating irrigation models on soil nutrients, soil salts and yield of tomato with fresh water (total dissolved solids of 0.50 g·L-1) and saline water (total dissolved solids of 3.01 g·L-1),…”, which contains the crop species and irrigation water salinity.
3. Line 30, the word “experimental” was added before “designed alternating irrigation models”, which show clearly that the “designed alternating irrigation models” was the six drip irrigation models as mentioned above.
4. Line 31, the analysis of tomato yield and fruit quality was added in the manuscript.
5. The phase “seedling stage” has been replaced by “seedling stage” in the whole document.
6. Line 39, the unsuitable expression “…HID (Hetao Irrigation District), however, with…” has been corrected to “…HID (Hetao Irrigation District). However, with…” in the Introduction part.
7. Line 48, the misspelling phase “garicultural irrigation” has been corrected to “agricultural irrigation” in the Introduction part.
8. Line 61, the misspelling word “applys” in line 48 has been corrected to “applies” in the Introduction part.
9. Line 99, the misspelling word “maximun” in line 99 has been corrected to “maximum” in the Introduction part.
10. Line 101, the redundant word “and” has been delated in the Introduction part.
11. Line 245, the misspelling word “damge” has been corrected to “damage” in the Results part.
12. Line 279, the misspelling word “minimun” has been corrected to “minimum” in the Results part.
13. Line 397, the redundant word “model” has been delated in the Conclusion part.
14. Line 412, the redundant word “to” has been delated in the Acknowledgments part.
15. Line 428, the misspelling word “cron” has been corrected to “corn” in the Reference part.
16. Line 541, the misspelling word “Ressource” has been corrected to “Resource” in the Reference part.
17. Line 90-91, the sentence “studies have shown that, transplantation can be adopted to alleviate the stress resistance of tomato to soil salt [26],” from line 90 to line 91 has been delated, as it has no sense to the study.
18. Line 103, the expression “for tomato cultivation” was added after “…based on growth stages under mulched drip irrigation”.
19. Line 104, the expression “in soil” was added after “…on total nutrient and available nutrient” in line 104.
20. Line 106, the unsuitable expression “…with different alternating irrigation models.” was replaced by “…with different water resources.” in line 106.
21. Line 126, the additional expression “(electrical conductivity of 0.79 dS·m-1)” has been added behind “…0.505 g·L-1” in Line 126, meanwhile, the additional expression “(electrical conductivity of 4.70 dS·m-1)” has been added behind “…3.006 g·L-1” in Line 126.
22. Line 131, the unsuitable expression “K++Na+” has been corrected to “K+ or Na+” in line 3 of Table 4; the unsuitable word “Project” has been corrected to “Characteristics” in the column names of Table 4.
23. Line 156, the expression “According to the extensive investigation in 2016,” was added before “The traditional tomato cultivation is generally planted by film covering,…” in Line 156.
24. Line 163, the expression “According to the traditional fertilization schedule, diammonium phosphate…” was replaced by “The fertilization schedule of present field experiment was based on the traditional fertilization schedule. Diammonium phosphate…” in Line 163.
25. Line 169, the unsuitable word “is” has been corrected to “are” in Line 169.
26. Line 157, the total irrigation quota for traditional surface flood irrigation during tomato growth period in HID is 3600 m3·ha-1, not 1200 m3·ha-1. As there are 3 times irrigation during tomato growth period, and the irrigation quota for each application is 1200 m3·ha-1, I have omitted the irrigation times.
27. Line 178, the additional expression “(May 13, 2017 and 2018)” has been added behind “…after the tomato seedlings grafting” in Line 178.
28. Line 150, the expression “each treatment was equipped with three vacuum gauge tensiometers which installed at 20 cm underneath the emitter for SMP monitoring” was added at the end of “2.2 Experimental design” in Line 150.
29. Line 179-180, the expression “…, as soon as the value of tensiometers lower than -25 kPa, irrigation was implemented with corresponding irrigation water resources,…” was replaced by “…, as soon as the value of all tensiometers in each treatment were lower than -25 kPa, irrigation was implemented for each treatment with corresponding irrigation water resources,…” in Line 179 and 180.
30. Line 140-142, there was some difference about the date for the beginning and ending of each growth period for all treatments, thus, the average beginning and ending date of each growth period was delated.
31. “2. Materials and Methods” part: “2.1 Experimental area” was renamed by “2.1 Experimental conditions”; “2.4 Observation and equipment” was renamed by “2.4 Measurements”; “2.4.1 Meteorological observation”, “2.4.2. SMP measurement” and “2.4.3. Soil basic properties” were delated, the original part of “2.4.1. Meteorological observation” and “2.4.3. Soil basic properties” were incorporated into “2.1 Experimental conditions”, while the former part of “2.4.2. SMP measurement” was incorporated into “2.2 Experimental design”.
32. The original “Figure 2” is placed in front of “Figure 1”.
33. The measurement of tomato yield was added in “2.4.3. Yield of tomato”,
34. Statistical error calculation and significance analysis has been added in Table 6, Table 7 and Table 8.
35. Figure 5(a) and Figure 5(b) has been updated, and the value has been added in Figure 5(a) and Figure 5(b).
36. Statistical error bars and letters for significance analysis has been added in Figure 6(a), Figure 6(b), Figure 6(c), Figure 6(d), Figure 8(a) and Figure 8(b).
37. Line 236-245, the result for Figure 5 was updated. “As detailed in Figure 5, the irrigation quota for all treatments at fruit-set stage and breaker stage was significantly higher than that at seeding stage and flowering stage. The consumed saline water for T4 treatment was 160mm in 2017 and 180mm in 2018, which was significantly higher than other treatments, however, the consumed fresh water for T4 treatment was the lowest. Both the irrigation quota for saline water and fresh water at flowering stage was 20mm, while at the same growth stage of tomato, the irrigation quota for brackish water was no less than that for fresh water. Moreover, the irrigation quota for T2 treatment was lower than other fresh-saline water alternating irrigation models, especially, the irrigation quota for T2 treatment was lower than that for fresh water irrigation. Additionally, continuous irrigation with saline water at seeding stage and flowering stage resulted in the decrease of irrigation quota at fruit-set stage and breaker stage.”
38. Line 239-242, the sentence “The result indicated that, saline water irrigation tends to improve the soil matric potential of root zone more easily than fresh water, especially at growth period with high water consumption such as fruit-set and breaker stage” in Line 239-242 was moved into the Discussion part.
39. Line 243-245, the sentence “which suggest that it was not suitable for continuous irrigation with saline water at seedling and flowering stage of tomato, the survival rate may be reduced and the tissues of plant may suffer from damage” was delated as no coherent.
40. Line 242, the expression “T4 treatment was more fresh water saving than other treatments” was added after the word “Moreover” in Line 242.
41. Line 251-252, the expression “which was consistent with the increasing of necessary nitrogen for plant tissues and fruit along the growth period” in Line 251-252 was moved into the Discussion part.
42. Line 253, the false sentence about the statistical analysis has been delated.
43. Line 258 the sentence was focused on the total phosphorus, the misspelled word “total nitrogen” has been replaced by “total phosphorus”.
44. Line 288 and Line 290-291 were moved to the Discussion part.
45. Line 288, the original expression “…,and the soil structure was hardly affected,…” was replaced by “…,and the soil structure for all treatments with brackish water irrigation was hardly affected,…”.
46. Line 303-305, the sentence from Line 303-305 has been moved into the Discussion.
47. Line 307-309, the expression “With the same designed threshold (-25 kPa) at each growth stage, the soil matric potential of the main root zone fluctuate more frequently for saline water irrigation than that for fresh water irrigation at the same growth period, which result in the quota of saline water usually higher than the amount of fresh water applied,…” was revised to “With the same designed threshold (-25 kPa) at each growth stage, the quota of saline water usually higher than the amount of fresh water applied,…”
48. Line 311-313, the sentence “Moreover, to avoid the damage of tomato plant tissues and keep considerable survival rate, continuous irrigation with saline water at seedling and flowering stage of tomato was not suitable” was delated.
49. Line 314-315, the unsuitable expression “Though the total nitrogen and the total phosphorus content in the root zone decreased gradually along with the tomato growth period, while…” was delated.
50. Line 321, the unsuitable expression “…then made impacts on…” was replaced by “… then affected the…” in Line 321.
51. Line 322-323, the sentence “Taking account of the nitrogen in the top soil, saline water irrigation was recommended before full bearing stage of tomato” was replaced by “Taking account of the nitrogen in the top soil, saline water irrigation was recommended before breaker stage of tomato (T2, T3, T6 treatment), because an increase of soil nitrogen content”.
52. Line 330-332, the sentence “Considering the soil phosphorus of the top soil for tomato plant, brackish water irrigation is not suggested at seeding stage of tomato” was delated.
53. Line 351, the unsuitable word “poison” was replaced by “toxic” in Line 351.
54. Line 352-352, the unsuitable sentence “With respect to the toxic function of Na+ to tomato roots, brackish water irrigation at tomato seedling stage (T2 and T6 treatment) is not recommended” was delated.
55. The Conclusion has been rewritten.
56. Line 242, the sentence “T4 treatment was more fresh water saving than other treatments” was added in Line 242.
57. Line 395, the expression “and beneficial for seed viability of tomato” was delated in Line 395.
58. Line 399, the expression “than other treatments” was added in Line 399, behind “…soil salt accumulation”.
59. Line 398, the expression “than T2, T3, T4 and T5 treatments” was added in Line 398.
60. Line 396, the unsuitable expression “, the damage of Cl- and Na+ to plant tissues,” in Line 396 was delated.
61. Line 395-397, the unsuitable expression “Considering …the alternating irrigation model for T3, T4 and T5 treatment was better than other treatments” has been replaced by “Considering …the alternating irrigation model for T2, T2 and T4 treatment was better than other treatments”.
62. The unsuitable expression “full bearing stage” has been replaced by “breaker stage” in the whole document.
63. Line 51, the original expression “…TDS (Total Dissolved Solids)…” has been replaced by “…total dissolved solids (TDS)…” in the Introduction part.
64. Line 105, the original expression “…SAR (Sodium Adsorption Ratio);” has been replaced by “…sodium adsorption ratio (SAR);” in the Introduction part.
65. Line 114, “Ding, Y.H.; Gao, X.Y.; Qu, Z.Y.; Jia, Y.L.; Hu, M.; Li, C.J. Effects of biochar application and irrigation methods on soil temperature in farmland. Water 2019, 11,449. doi: 10.3390/w11030499” was added as the citation of evaporation in Line114.
66. Line 174, the original expression “… the SMP (soil matric potential)…” has been replaced by “…the soil matric potential (SMP)…” in the Materials and Methods part.
67. The misspelling word “damge” has been corrected to “damage” in the whole document.
68. The analysis of tomato production and fruit quality (total soluble solids (TSS), lycopene, total sugar (Ts) and total acid (Ta)) were added in the manuscript.
69. The reference “Ding, Y.H.; Gao, X.Y.; Qu, Z.Y.; Jia, Y.L.; Hu, M.; Li, C.J. Effects of biochar application and irrigation methods on soil temperature in farmland. Water 2019, 11,449. doi: 10.3390/w11030499” has been added to the Reference part.
70. The reference “Brandt, S.; Pék, Z.; Barna, É.; Lugasi, A.; Helyes, L. Lycopene content and colour of ripening tomatoes as affected by environmental conditions. J. Sci. Food Agric. 2006, 86(4), 568-572. doi:10.1002/jsfa.2390” has been added to the Reference part.
71. The reference “Abdel Gawad, G.; Arslan, A.; Gaihbe, A.; Kadouri, F. The effects of saline irrigation water management and salt tolerant tomato varieties on sustainable production of tomato in Syria (1999-2002). Agricultural Water Management 2005, 78(1-2), 39-53. doi: 10.1016/j.agwat.2005.04.024” has been added to the Reference part.
72. Line 428, the misspelling word “cron” has been corrected to “corn” in the Reference part.
73. Line 472, the misspelling word “elaeagnus” has been corrected to “Elaeagnus” in the Reference part.
74. Line 480, the “doi: 10.1016/j.agwat.2005.04.016” has been added to the twenty-third reference in the Reference part.
75. Line 541, the misspelling word “Ressource” has been corrected to “Resource” in the Reference part.
Once again, thank you very much for your kind help.
Best regards,
Sincerely,

Reviewer 3 Report
Review of Water-552036
Effects of Alternating Irrigation on the Soil Salt and Soil Nutrient with Fresh-Saline Water
GENERAL COMMENTS
It is a remarkable manuscript, which is acceptable with minor revision.
Instead of “full bearing stage” term “breaker” is used more often, e.g. Brandt et al., 2006 (see references), please change it in the whole document.
Satistical error calculation is missing in all figures and tables.
INTRODUCTION
L51: total dissolved solids (TDS)
L105: “…sodium adsorption rate (SAR)…
MATERIALS AND METHODS
L110-114: Missing citation of evaporation, please complete.
L174: …soil matric potential (SMP)…
RESULTS AND DISCUSSION
L245: damage.
Conclusion
It is difficult to interpret results, without yield and its parameters (e.g. Brix).
REFERENCES
Please complete missing references:
Brandt, S. , Pék, Z. , Barna, É. , Lugasi, A. and Helyes, L. (2006), Lycopene content and colour of ripening tomatoes as affected by environmental conditions. J. Sci. Food Agric., 86: 568-572. doi:10.1002/jsfa.2390
Author Response
Dear Reviewer,
Thank you very much for your careful review of our manuscript, we appreciate you very much for the suggestions.
I have revised the manuscript carefully according your suggestions during the past seven days.
The main corrections in the manuscript are as flowing:
1. The unsuitable expression “full bearing stage” has been replaced by “breaker stage” in the whole document.
2. Statistical error calculation and significance analysis has been added in Table 6, Table 7 and Table 8.
3. Figure 5(a) and Figure 5(b) has been updated, and the value has been added in Figure 5(a) and Figure 5(b).
4. Statistical error bars and letters for significance analysis has been added in Figure 6(a), Figure 6(b), Figure 6(c), Figure 6(d), Figure 8(a) and Figure 8(b).
5. Line 51, the original expression “…TDS (Total Dissolved Solids)…” has been replaced by “…total dissolved solids (TDS)…” in the Introduction part.
6. Line 105, the original expression “…SAR (Sodium Adsorption Ratio);” has been replaced by “…sodium adsorption ratio (SAR);” in the Introduction part.
7. Line 114, “Ding, Y.H.; Gao, X.Y.; Qu, Z.Y.; Jia, Y.L.; Hu, M.; Li, C.J. Effects of biochar application and irrigation methods on soil temperature in farmland. Water 2019, 11,449. doi: 10.3390/w11030499” was added as the citation of evaporation in Line114.
8. Line 174, the original expression “… the SMP (soil matric potential)…” has been replaced by “…the soil matric potential (SMP)…” in the Materials and Methods part.
9. The misspelling word “damge” has been corrected to “damage” in the whole document.
10. The analysis of tomato production and fruit quality (total soluble solids (TSS), lycopene, total sugar (Ts) and total acid (Ta)) were added in the manuscript.
11.
12. The reference “Ding, Y.H.; Gao, X.Y.; Qu, Z.Y.; Jia, Y.L.; Hu, M.; Li, C.J. Effects of biochar application and irrigation methods on soil temperature in farmland. Water 2019, 11,449. doi: 10.3390/w11030499” has been added to the Reference part.
13. The reference “Brandt, S.; Pék, Z.; Barna, É.; Lugasi, A.; Helyes, L. Lycopene content and colour of ripening tomatoes as affected by environmental conditions. J. Sci. Food Agric. 2006, 86(4), 568-572. doi:10.1002/jsfa.2390” has been added to the Reference part.
14. The reference “Abdel Gawad, G.; Arslan, A.; Gaihbe, A.; Kadouri, F. The effects of saline irrigation water management and salt tolerant tomato varieties on sustainable production of tomato in Syria (1999-2002). Agricultural Water Management 2005, 78(1-2), 39-53. doi: 10.1016/j.agwat.2005.04.024” has been added to the Reference part.
15. Line 428, the misspelling word “cron” has been corrected to “corn” in the Reference part.
16. Line 472, the misspelling word “elaeagnus” has been corrected to “Elaeagnus” in the Reference part.
17. Line 480, the “doi: 10.1016/j.agwat.2005.04.016” has been added to the twenty-third reference in the Reference part.
18. Line 541, the misspelling word “Ressource” has been corrected to “Resource” in the Reference part.
19. Line 3, the expression “Yield of Tomato” was added into the Title.
20. Line 17-18, the sentence “To study the effect of alternating irrigation models on soil nutrients and soil salts,…” was replaced by “To study the effect of alternating irrigation models on soil nutrients, soil salts and yield of tomato with fresh water (total dissolved solids of 0.50 g·L-1) and saline water (total dissolved solids of 3.01 g·L-1),…”, which contains the crop species and irrigation water salinity.
21. Line 30, the word “experimental” was added before “designed alternating irrigation models”, which show clearly that the “designed alternating irrigation models” was the six drip irrigation models as mentioned above.
22. The phase “seedling stage” has been replaced by “seedling stage” in the whole document.
23. Line 39, the unsuitable expression “…HID (Hetao Irrigation District), however, with…” has been corrected to “…HID (Hetao Irrigation District). However, with…” in the Introduction part.
24. Line 48, the misspelling phase “garicultural irrigation” has been corrected to “agricultural irrigation” in the Introduction part.
25. Line 61, the misspelling word “applys” in line 48 has been corrected to “applies” in the Introduction part.
26. Line 99, the misspelling word “maximun” in line 99 has been corrected to “maximum” in the Introduction part.
27. Line 101, the redundant word “and” has been delated in the Introduction part.
28. Line 245, the misspelling word “damge” has been corrected to “damage” in the Results part.
29. Line 279, the misspelling word “minimun” has been corrected to “minimum” in the Results part.
30. Line 397, the redundant word “model” has been delated in the Conclusion part.
31. Line 412, the redundant word “to” has been delated in the Acknowledgments part.
32. Line 428, the misspelling word “cron” has been corrected to “corn” in the Reference part.
33. Line 541, the misspelling word “Ressource” has been corrected to “Resource” in the Reference part.
34. Line 90-91, the sentence “studies have shown that, transplantation can be adopted to alleviate the stress resistance of tomato to soil salt [26],” from line 90 to line 91 has been delated, as it has no sense to the study.
35. Line 103, the expression “for tomato cultivation” was added after “…based on growth stages under mulched drip irrigation”.
36. Line 104, the expression “in soil” was added after “…on total nutrient and available nutrient” in line 104.
37. Line 106, the unsuitable expression “…with different alternating irrigation models.” was replaced by “…with different water resources.” in line 106.
38. Line 126, the additional expression “(electrical conductivity of 0.79 dS·m-1)” has been added behind “…0.505 g·L-1” in Line 126, meanwhile, the additional expression “(electrical conductivity of 4.70 dS·m-1)” has been added behind “…3.006 g·L-1” in Line 126.
39. Line 131, the unsuitable expression “K++Na+” has been corrected to “K+ or Na+” in line 3 of Table 4; the unsuitable word “Project” has been corrected to “Characteristics” in the column names of Table 4.
40. Line 156, the expression “According to the extensive investigation in 2016,” was added before “The traditional tomato cultivation is generally planted by film covering,…” in Line 156.
41. Line 163, the expression “According to the traditional fertilization schedule, diammonium phosphate…” was replaced by “The fertilization schedule of present field experiment was based on the traditional fertilization schedule. Diammonium phosphate…” in Line 163.
42. Line 169, the unsuitable word “is” has been corrected to “are” in Line 169.
43. Line 157, the total irrigation quota for traditional surface flood irrigation during tomato growth period in HID is 3600 m3·ha-1, not 1200 m3·ha-1. As there are 3 times irrigation during tomato growth period, and the irrigation quota for each application is 1200 m3·ha-1, I have omitted the irrigation times.
44. Line 178, the additional expression “(May 13, 2017 and 2018)” has been added behind “…after the tomato seedlings grafting” in Line 178.
45. Line 150, the expression “each treatment was equipped with three vacuum gauge tensiometers which installed at 20 cm underneath the emitter for SMP monitoring” was added at the end of “2.2 Experimental design” in Line 150.
46. Line 179-180, the expression “…, as soon as the value of tensiometers lower than -25 kPa, irrigation was implemented with corresponding irrigation water resources,…” was replaced by “…, as soon as the value of all tensiometers in each treatment were lower than -25 kPa, irrigation was implemented for each treatment with corresponding irrigation water resources,…” in Line 179 and 180.
47. Line 140-142, there was some difference about the date for the beginning and ending of each growth period for all treatments, thus, the average beginning and ending date of each growth period was delated.
48. “2. Materials and Methods” part: “2.1 Experimental area” was renamed by “2.1 Experimental conditions”; “2.4 Observation and equipment” was renamed by “2.4 Measurements”; “2.4.1 Meteorological observation”, “2.4.2. SMP measurement” and “2.4.3. Soil basic properties” were delated, the original part of “2.4.1. Meteorological observation” and “2.4.3. Soil basic properties” were incorporated into “2.1 Experimental conditions”, while the former part of “2.4.2. SMP measurement” was incorporated into “2.2 Experimental design”.
49. The original “Figure 2” is placed in front of “Figure 1”.
50. The measurement of tomato yield was added in “2.4.3. Yield of tomato”,
51. Line 236-245, the result for Figure 5 was updated. “As detailed in Figure 5, the irrigation quota for all treatments at fruit-set stage and breaker stage was significantly higher than that at seeding stage and flowering stage. The consumed saline water for T4 treatment was 160mm in 2017 and 180mm in 2018, which was significantly higher than other treatments, however, the consumed fresh water for T4 treatment was the lowest. Both the irrigation quota for saline water and fresh water at flowering stage was 20mm, while at the same growth stage of tomato, the irrigation quota for brackish water was no less than that for fresh water. Moreover, the irrigation quota for T2 treatment was lower than other fresh-saline water alternating irrigation models, especially, the irrigation quota for T2 treatment was lower than that for fresh water irrigation. Additionally, continuous irrigation with saline water at seeding stage and flowering stage resulted in the decrease of irrigation quota at fruit-set stage and breaker stage.”
52. Line 239-242, the sentence “The result indicated that, saline water irrigation tends to improve the soil matric potential of root zone more easily than fresh water, especially at growth period with high water consumption such as fruit-set and breaker stage” in Line 239-242 was moved into the Discussion part.
53. Line 243-245, the sentence “which suggest that it was not suitable for continuous irrigation with saline water at seedling and flowering stage of tomato, the survival rate may be reduced and the tissues of plant may suffer from damage” was delated as no coherent.
54. Line 242, the expression “T4 treatment was more fresh water saving than other treatments” was added after the word “Moreover” in Line 242.
55. Line 251-252, the expression “which was consistent with the increasing of necessary nitrogen for plant tissues and fruit along the growth period” in Line 251-252 was moved into the Discussion part.
56. Line 253, the false sentence about the statistical analysis has been delated.
57. Line 258 the sentence was focused on the total phosphorus, the misspelled word “total nitrogen” has been replaced by “total phosphorus”.
58. Line 288 and Line 290-291 were moved to the Discussion part.
59. Line 288, the original expression “…,and the soil structure was hardly affected,…” was replaced by “…,and the soil structure for all treatments with brackish water irrigation was hardly affected,…”.
60. Soil samples were taken during each growth period at depth of 0-40 cm and 40-100 cm in the two locations which 0 cm (inside the film) and 75 cm (outside the film) from the drip tube for the measurement of soil salinity.
61. Line 303-305, the sentence from Line 303-305 has been moved into the Discussion.
62. Line 307-309, the expression “With the same designed threshold (-25 kPa) at each growth stage, the soil matric potential of the main root zone fluctuate more frequently for saline water irrigation than that for fresh water irrigation at the same growth period, which result in the quota of saline water usually higher than the amount of fresh water applied,…” was revised to “With the same designed threshold (-25 kPa) at each growth stage, the quota of saline water usually higher than the amount of fresh water applied,…”
63. Line 311-313, the sentence “Moreover, to avoid the damage of tomato plant tissues and keep considerable survival rate, continuous irrigation with saline water at seedling and flowering stage of tomato was not suitable” was delated.
64. Line 314-315, the unsuitable expression “Though the total nitrogen and the total phosphorus content in the root zone decreased gradually along with the tomato growth period, while…” was delated.
Statistical error bars and letters for significance analysis has been added in Figure 6(a) and Figure 6(c).
65. Line 321, the unsuitable expression “…then made impacts on…” was replaced by “… then affected the…” in Line 321.
66. Line 322-323, the sentence “Taking account of the nitrogen in the top soil, saline water irrigation was recommended before full bearing stage of tomato” was replaced by “Taking account of the nitrogen in the top soil, saline water irrigation was recommended before breaker stage of tomato (T2, T3, T6 treatment), because an increase of soil nitrogen content”.
67. Line 330-332, the sentence “Considering the soil phosphorus of the top soil for tomato plant, brackish water irrigation is not suggested at seeding stage of tomato” was delated.
68. Line 351, the unsuitable word “poison” was replaced by “toxic” in Line 351.
69. Line 352-354, the unsuitable sentence “With respect to the toxic function of Na+ to tomato roots, brackish water irrigation at tomato seedling stage (T2 and T6 treatment) is not recommended” was delated.
70. The Conclusion has been rewritten.
71. Line 242, the sentence “T4 treatment was more fresh water saving than other treatments” was added in Line 242.
72. Line 395, the expression “and beneficial for seed viability of tomato” was delated in Line 395.
73. Line 399, the expression “than other treatments” was added in Line 399, behind “…soil salt accumulation”. The soil salt content for T2 treatment in 0-2500px was lower than other alternating irrigation treatments (Figure 8).
74. Line 398, the expression “than T2, T3, T4 and T5 treatments” was added in Line 398.
75. Line 396, the unsuitable expression “, the damage of Cl- and Na+ to plant tissues,” in Line 396 was delated.
76. Line 395-397, the unsuitable expression “Considering …the alternating irrigation model for T3, T4 and T5 treatment was better than other treatments” has been replaced by “Considering …the alternating irrigation model for T2, T2 and T4 treatment was better than other treatments”.
Once again, thank you very much for your review.
Best regards,
Sincerely,
2. Materials and Methods
2.4.3. Fruit yield and quality
Tomato red fruits were picked twice a week during harvest seasons for each plot, and all fruits were classified as unmarketable (fruits which were cracked, green, sunburn, with symptoms of blossom-end rot, or damaged by pests) or marketable ones, the fruit yield was measured for the marketable fruits of each plot. The yield (g/plant) on each plant were determined during the harvest season.
For each treatment, 20 remarkable tomato fruits were collected randomly for the measurement of total soluble solids (TSS), lycopene, total sugar (Ts) and total acid (Ta). The TSS was measured by the ACT-1E digital refractometer (ATAGO company, Japan), and the content of Lycopene was measured using the spectrophotometry method. The Ts was determined by the Fehling reagent titration method, and the Ta was measured by the sodium hydroxide titration method.
3. Results
3.6 Tomato production
Compared with fresh water irrigation, saline water irrigation has dual effects on plant, on the one hand, more salt was carried into the top soil which was not conducive to root water uptake, on the other hand, some salt ions of brackish water are benefit for the vegetative and reproductive growth of tomato. The remarkable fruit yield of tomato is listed in Table 7.
Table 7. Remarkable tomato fruit yield
Treatment | Year 2017 | Year 2018 | ||
Fruit yield on each plant (g/plant) | Tomato production (104kg·ha-1) | Fruit yield on each plant (g/plant) | Tomato production (104kg·ha-1) | |
T1 | 45±5c | 9.565±0.326c | 43±4c | 9.443±0.423d |
T2 | 29±3a | 7.821±0.345a | 32±3a | 8.012±0.541a |
T3 | 36±4b | 8.646±0.432b | 35±2ab | 8.632±0.318c |
T4 | 40±4b | 9.302±0.517c | 38±2b | 9.264±0.426d |
T5 | 37±3b | 8.543±0.429b | 36±3ab | 8.412±0.445bc |
T6 | 37±2b | 8.305±1.338b | 38±5b | 8.226±0.439ab |
Note: Values in a column followed by the same letter are not significantly different at p ≤ 0.05.
According to Table 7, the tomato production for treatments was in the order of T1>T4>T3>T5>T6>T2, and the remarkable fruit yield on each plant for T4 treatment was the highest among alternating irrigation models both in 2017 and 2018. Compared with T1 treatment, the tomato production for T4 treatment just decreased by 2.75% in 2017 and 1.90% in 2018, respectively, while the tomato production for T2 treatment decreased by 18.2% in 2017 and 15.15% in 2018, respectively.
3.7 Fruit quality
To estimate the fruit quality, the common parameters (TSS, Lycopene, Ts and Ta) were measured after tomato harvest, and the results are shown in Table 8.
Table 8. Fruit quality parameters of tomato
Treatments | Year 2017 | Year 2018 | ||||||
TSS (%) | Lycopene (mg/100g) | TS (%) | Ta (%) | TSS (%) | Lycopene (mg/100g) | TS (%) | Ta (%) | |
T1 | 7.94±1.23a | 6.50±0.54a | 7.12±0.82a | 1.54±0.06b | 8.04±1.02a | 6.54±0.24a | 7.24±0.32a | 1.47±0.25ab |
T2 | 8.04±2.45a | 6.92±0.43b | 7.80±0.35c | 1.30±0.04ab | 8.02±1.14a | 7.05±0.26b | 7.76±0.43b | 1.28±0.13b |
T3 | 8.11±2.34a | 7.12±1.05b | 7.90±0.76cd | 1.23±0.24ab | 8.18±0.36a | 7.08±0.18b | 7.88±0.56bc | 1.31±0.24ab |
T4 | 8.30±3.04a | 7.70±0.68c | 8.20±1.12d | 0.98±0.17a | 8.34±0.85a | 7.53±0.19c | 8.17±0.78c | 1.04±0.16a |
T5 | 8.02±2.51a | 7.22±1.24b | 7.94±0.72cd | 1.32±0.34ab | 8.12±0.73a | 7.11±0.12b | 8.02±0.49bc | 1.35±0.34ab |
T6 | 8.18±2.25a | 7.05±0.57b | 7.46±0.46b | 1.22±0.22ab | 8.20±0.48a | 6.98±0.07b | 7.38±0.67a | 1.18±0.27ab |
Note: Values in a column followed by the same letter are not significantly different at p ≤ 0.05.
As shown in Table 8, the TSS, lycopene and TS of tomato fruit for fresh-saline water alternating irrigation was higher than that for fresh water irrigation at all growth stages, moreover, the TSS, lycopene and TS for T4 treatment was the highest among treatments. Additionally, the Ta for fresh-saline water alternating irrigation was lower than that for fresh water irrigation, and the T4 treatment reached the minimum value.
4. Discussion
Additionally, compared with fresh water irrigation, the tomato fruit yield for fresh-saline water alternating irrigation models decreased by 2.75%-18.20% in 2017 and 1.90%-15.15% in 2018, respectively, which indicated that, fresh-saline water alternating irrigation will result in the reduction of tomato production, but with proper irrigation models, the reduction can be controlled within a certain degree. Both the TSS, lycopene and TS of tomato fruit for T2, T3, T4, T5 and T6 treatment was higher than that for T1 treatment, while the Ta for fresh water irrigation was higher than fresh-saline water alternating irrigation, which indicated that, the trace elements of brackish water is benefit for the compounding of TSS, lycopene and TS, which result was identical to the study of Abdel Gawad, who reported that the soluble solid content of tomato fruit raised with a certain amount of brackish water [56].

Round 2
Reviewer 1 Report
I reviewed the first version of this manuscript and I recommended that the paper be rejected . Now, the manuscript has improved a lot. Authors have taken into account most of the comments and suggestions. Therefore, my recommendation is that the revised manuscript is acceptable for publication.